# BeanCounter: A low-toxicity, large-scale, and open dataset of business-oriented text

**Siyan Wang**
University of Chicago
Chicago, IL 60637

**Bradford Levy**[*]
University of Chicago
Chicago, IL 60637

## Abstract

Many of the recent breakthroughs in language modeling have resulted from scaling effectively the same model architecture to larger datasets. In this vein, recent work has highlighted performance gains from increasing training dataset size and quality, suggesting a need for novel sources of large-scale datasets. In this work, we introduce BeanCounter, a public dataset consisting of more than 159B tokens extracted from businesses' disclosures. We show that this data is indeed novel: less than 0.1% of BeanCounter appears in Common Crawl-based datasets and the data is an order of magnitude larger than datasets relying on similar sources. Given the data's provenance, we hypothesize that BeanCounter is comparatively more factual and less toxic than web-based datasets. Exploring this hypothesis, we find that many demographic identities occur with similar prevalence in BeanCounter but with *significantly less toxic* context relative to other datasets. To demonstrate the utility of BeanCounter, we evaluate and compare two LLMs continually pre-trained on BeanCounter with their base models. We find an 18-33% reduction in toxic generation and improved performance within the finance domain for the continually pretrained models. Collectively, our work suggests that BeanCounter is a novel source of low-toxicity and high-quality domain-specific data with sufficient scale to train multi-billion parameter LLMs.

## 1 Introduction

A key ingredient to the recent breakthroughs in language modeling has been the availability of large-scale datasets. Along these lines, recent work has shown that training incrementally larger language models demands a similar increase in training data [29, 27], and that data quality is a significant determinant of a model's ultimate performance [38, 23]. However, the creation and scaling of text datasets sourced from the public domain raises a variety of concerns such as inclusion of personally identifiable information [30], degrading quality [32], and biases or false information [37].

In this work, we contribute to overcoming the challenges of scaling text datasets. Specifically, we introduce BeanCounter, a dataset comprised of more than 159B tokens extracted from public domain business-oriented disclosures. The introduction of BeanCounter makes four primary contributions:

**1. Novel large-scale dataset of domain specific content.** BeanCounter consists of content produced by businesses to communicate with a variety of stakeholders such as investors and regulators. This content is not easily accessible via web scraping and we find that little of BeanCounter is included in other commonly used datasets. For example, less than 0.1% of BeanCounter is in C4 [16]. In

---

[*]Corresponding author: `bradford.levy@chicagobooth.edu`
Code: `https://github.com/bradfordlynch/beancounter`
Data: `https://huggingface.co/datasets/bradfordlevy/BeanCounter`

considering the scale of BeanCounter, we define "large-scale" as sufficient size to pre-train a multi-billion parameter model [10], and of similar order of magnitude as other web-based datasets (i.e., 100B+ tokens) [16]. Along these lines, BeanCounter consists of more than 159B tokens of cleaned text and more than 111B tokens of deduplicated text (see Table 1). Thus, to the best of our knowledge, BeanCounter is the largest and most comprehensive public dataset of business-oriented text.

Table 1: Statistics for the BeanCounter duplication rate

| Dataset | # tokens | size | Form type | # duplicate tokens | % duplicate tokens |
|---|---|---|---|---|---|
| BeanCounter.clean | 159.4B | 865.2 GB | 10-K | 1.0B | 8.78% |
| BeanCounter.fraud | 0.3B | 1.7 GB | 10-Q | 1.1B | 8.80% |
| BeanCounter.final | 110.5B | 598.8 GB | 8-K | 3.9B | 23.3% |

(a) BeanCounter.final is cleaned and deduplicated on document level.   (b) The most commonly studied form types and the duplication rate in BeanCounter.

**2. Timely, factual, and high-quality content.** Considering timeliness, Dhingra et al. [15] show that incorporating the concept of time into LLMs can enhance their ability to recall time-dependent facts, e.g., the current president of the United States. In contrast to web-based datasets which usually do not have a precise timestamp of when the web page was created or updated, every observation in BeanCounter has a timestamp of when the content became available–accurate to the second. Considering factuality and quality, there are at least two reasons BeanCounter is comparatively more factual and of higher-quality than web-based datasets. First, the principal executive and financial officers, e.g., the CEO and CFO, must certify the disclosures from which BeanCounter is sourced [48]. Second, these disclosures are the primary channel used by businesses to communicate with stakeholders. Thus, the businesses are incentivized to produce effective communications. In this sense, BeanCounter supports future research on the role of time in LLMs, factuality, and quality.

**3. Toxicity and demographic identity analysis.** As access to LLMs has proliferated, a common concern is the extent to which LLMs produce toxic or otherwise harmful content. Along these lines, a large literature has examined the toxicity of various web-based datasets. In this vein, we examine the prevalence of a variety of demographic identity descriptors and the toxicity of text in which they are mentioned [3, 50]. We contrast our findings to text from C4 [16] and find that many descriptors are similarly prevalent in BeanCounter but the context in which they are mentioned is *significantly less toxic*. For instance, the descriptor "Asian" is mentioned in 4.67% and 9.26% of documents in C4 and filings in BeanCounter, respectively, yet the text surrounding "Asian" in BeanCounter is 77.0% less toxic, on average. Similarly, "LGBTQ" is mentioned 0.37% and 0.17% in C4 and BeanCounter, respectively, yet the toxicity of its surrounding content is 85.7% lower in BeanCounter.[1] We find this general trend across nearly all descriptors.

**4. Model Evaluation.** Given the influence of training data on model generation, our analyses suggest that models trained on BeanCounter may exhibit less toxic generation. We explore this possibility by continuing the pre-training of models on BeanCounter and evaluate their performance on domain-specific tasks and toxic generation. We find that models trained on BeanCounter exhibit better performance within financial applications while exhibiting 18-33% less toxic generation.

In summary, this paper introduces a novel business-oriented text dataset comprised of more than 159B tokens which are comparatively more factual, of higher-quality, and less toxic than commonly used web-based datasets. We have open-sourced BeanCounter and made it available via Hugging Face Hub. We hope that BeanCounter will enable the creation of new foundation models which are less likely to generate toxic content and are better suited to business-oriented tasks.

## 2  Related Work

**Foundation models and large-scale text datasets.** The current scale of Transformer-based language models has grown more than 1000-fold since early examples of such models, e.g., from GPT-1

---

[1]A table with the most prevalent descriptors in C4-en is shown in Appendix C.

[43] to OPT-175B [63]. While the scale of these models has grown dramatically, the size of the data has grown comparatively more slowly, e.g., even though OPT-175B is 100 times larger than GPT-2 [44] it was trained on only 10 times as much data (180B versus roughly 20B tokens). In this vein, recent work has explored how to best allocate a fixed *training* compute budget and found results which suggest earlier models were significantly under-trained, i.e., a 10-fold increase in model size should correspond to a 10-fold increase in training data size [29, 27]. Alternatively, one can view this resource allocation problem as desiring the best possible *inference* performance for a fixed budget [54]. In which case, models should be trained on even more data than that suggested by the scaling laws of Hoffmann et al. [27]. The common theme throughout this literature is that an increase in model size requires at least a similar increase in training data as well. This begs the question of where to source the additional data.

Early models such as BERT [14] and GPT-1 [43] sourced training data from books [65] and/or Wikipedia. As models scaled, researchers began collecting larger datasets by scraping text from the web, e.g., WebText consists of upvoted links from Reddit [44], and this approach is now the dominant method for assembling large-scale text datasets [21, 29, 26, 61, 46, 18, 16, 34, 54, 42]. While this is now the dominant approach, it is not without concerns. These concerns range from socially benign, e.g., low-quality content that leads to gibberish generations [56] or large amounts of duplicate content [32], to more socially harmful outcomes such as generation which is: biased and/or toxic [49, 59, 20, 35], revealing of personally identifiable information [30], or factually inaccurate [37, 41]. In this paper, we present a novel dataset sourced from business-oriented content which is more factually accurate, less toxic, and paired with the time the content became public.

**Business-oriented text datasets.** Early business-oriented corpuses for NLP research were generally of small scale, i.e., tens of millions of tokens [33, 40, 5], and therefore more suitable for fine-tuning or downstream evaluation than pre-training. Recently, datasets on the scale of billions of tokens have been developed using proprietary data, e.g., Bloomberg's catalog of content [62], and publicly available data, e.g., content from the Securities and Exchange Commission's (SEC) Electronic Data Gathering and Retrieval (EDGAR) system. Notably, prior work using EDGAR data generally considers only a subset of the disclosures posted to EDGAR. For example, Loukas et al. [39] extract text solely from annual reports to create a dataset of 6.5B tokens. Wu et al. [62] extract text from annual *and* quarterly reports to create a *non*-public dataset of 14.5B tokens. Other datasets relying on EDGAR are of significantly smaller size, i.e., less than 1B tokens [31, 57]. We extend this literature by considering all disclosures on EDGAR and find that 83% of BeanCounter comes from disclosure types not included in prior work. Additionally, we consider not only the main content of EDGAR disclosures but attachments as well and find that 29% of tokens are sourced from attachments.[2]

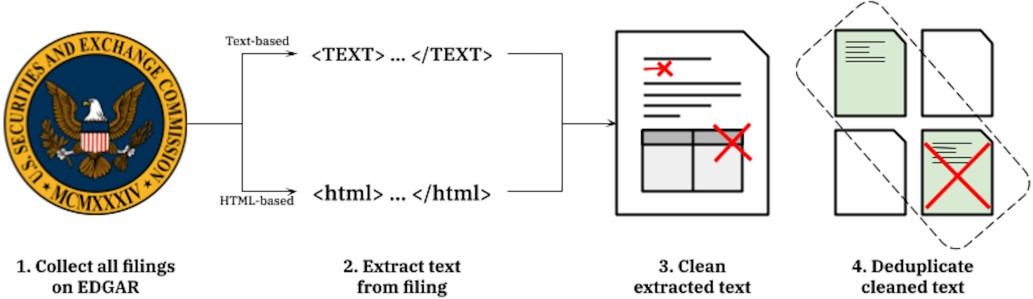

Figure 1: **Overview of dataset construction:** All EDGAR filings are downloaded, text is extracted from filings using content-type specific extractors, extracted text is then cleaned and deduplicated.

---

[2]For example, consider American Airlines' 2016 Annual Report available from EDGAR here and in the dataset of Loukas et al. [39] under the filename "6201_2016.htm." In addition to the main document (Type 10-K on EDGAR), American Airlines also attaches a credit agreement (Type EX-10.1 on EDGAR). The credit agreement contains a similar amount of text as the main 10-K document but does not appear in Loukas et al. [39]. We find that more than 89% of annual reports include at least one such attachment.

# 3 Dataset Construction and Analysis

BeanCounter is constructed from all public filing submissions to EDGAR–the main mechanism through which entities satisfy their disclosure obligations under the Securities Acts of 1933 and 1934, the Trust Indenture Act of 1939, and the Investment Company Act of 1940 [1]. Professional and amateur investors often rely on SEC filings to make informed investment decisions. The SEC allows unrestricted copying and redistribution of the content on EDGAR [4]. Notably, entities which file false or misleading information to EDGAR are subject to legal jeopardy since harmed individuals can seek recourse against them. Along these lines, when a firm does engage in fraud and is caught, the SEC generally publishes an enforcement action against the firm. In creating BeanCounter, we leverage these enforcement actions to filter content produced by known fraudulent firms into a distinct split of the dataset. In this sense, EDGAR serves as one of the largest public repositories of business-oriented content that is more likely to be factual than general web content or literary works. While BeanCounter includes personal identifiable information such as names and phone numbers, the entities have consented to making this information publicly available and the SEC allows for confidential treatment requests if they would prefer the information be withheld.

A flowchart representing our dataset pipeline is presented in Figure 1. At a high-level, our pipeline involves four steps: (1) collection of all filings on EDGAR, (2) extraction of text from those filings, (3) cleaning of extracted text, and (4) deduplication of cleaned text. For each step, we follow the relevant best practices developed in recent years. The full details of dataset construction can be found in Appendix A. The resulting cleaned, fraud, and deduplicated datasets contain roughly 159B , 308M and 111B  tokens, respectively (see Table 1 for more details).

Prior literature has highlighted the sensitivity of downstream model performance on dataset content, e.g., Li et al. [35] finds that existing biases in the model can be amplified by fine-tuning. Following previous work [45, 55], we analyze the content of BeanCounter along a variety of dimensions including content source (e.g., industry, firm, and form type), demographic representation, and toxicity to understand the implications of either pre-training or fine-tuning on BeanCounter.

## 3.1 Content Analysis

**Year and Form Type.**   Figure 2 presents the volume of content in BeanCounter by year and form-type. As illustrated in Figure 2a, there are roughly 3.9B tokens per year on average and there is a general increase in number of tokens filed throughout time. Considering the source of content by form type, Figure 2b shows that four form types contribute 43.7% of all tokens in BeanCounter: 485BPOS (separate account registration statement for investment companies), 8-K (report of unscheduled material events or corporate changes), 10-Q (quarterly report) and 10-K (annual report). In Appendix D we present examples of content from the ten most common form types in BeanCounter. The content is surprisingly diverse, ranging from discussions of financial performance (see Figure D.2) to Amazon.com's plans to use IPO funds for the implementation of collaborative filtering [22] (see Figure D.9).

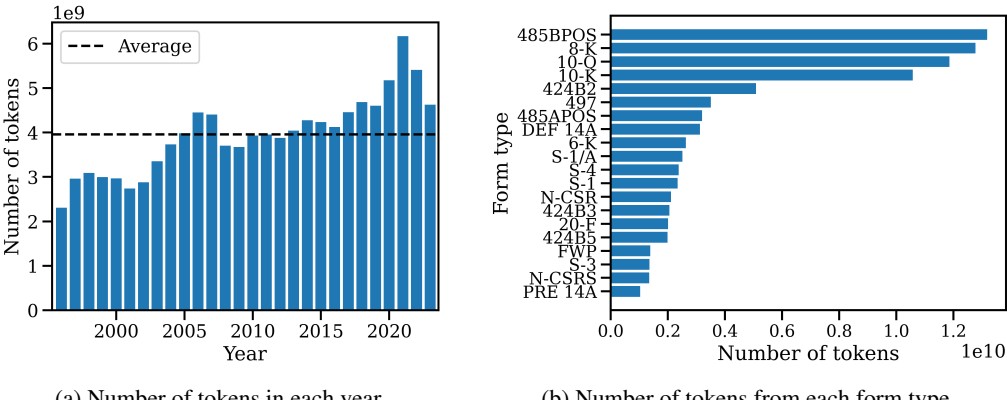

(a) Number of tokens in each year.          (b) Number of tokens from each form type.

Figure 2: Text volume by year and form type.

Compared to prior literature, we find that our approach appears to extract nearly twice as much content: 10.9B vs. 6.5B tokens from annual reports [39] and 22.6B vs. 14.5B tokens from both annual and quarterly reports [62]. Based on manual inspection of Loukas et al. [39], we attribute this difference to our processing of not only the main filing but also any attachments.

**Industry and Firm.** To better understand group representation in BeanCounter, we explore which industries and firms contribute the most content. We adopt the commonly used industry classification of Fama and French [17] to assign each firm to one of 48 high-level industries. Figure 3 shows that Banks and Financial Services are the two most represented industries followed by Business Services, Pharmaceuticals, and Chip Fabrication.

Next, we consider which specific firms are most prevalent in BeanCounter. After matching on firm identification from the S&P Capital IQ dataset, there are 16.6K firms in the BeanCounter dataset. Given that the financial services industries are very highly represented, we present representation separately for the top 20 firms and the top 20 *non-financial* firms, i.e. all firms not in the Finance, Insurance, Banks nor Real Estate industries. Figure 4a shows that many well-known financial firms, e.g., Goldman Sachs, contribute large volumes of content to BeanCounter. Figure 4b shows that the most represented non-financial firms tend to be older, well-established firms such as AT&T and United Airlines.

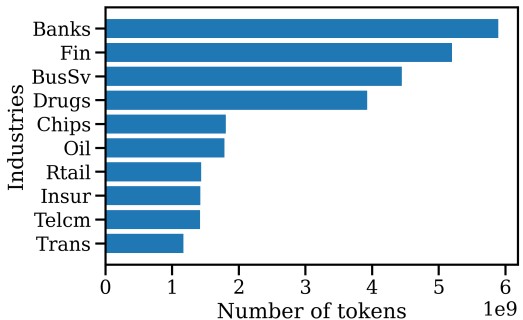

Figure 3: Top 10 industries with the highest token volume.

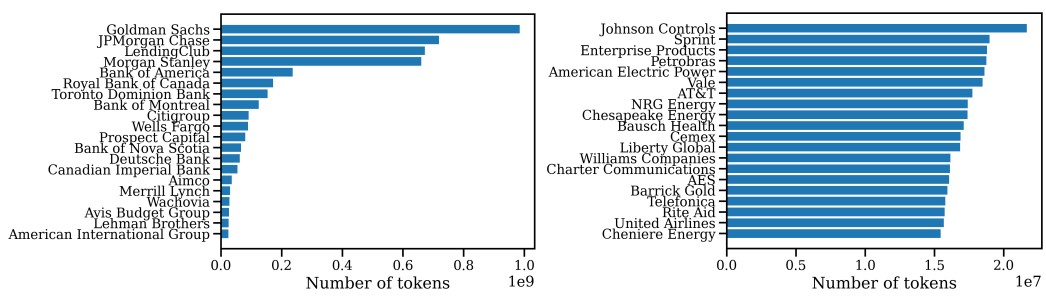

(a) Top 20 contributing firms.          (b) Top 20 non-financial contributing firms.

Figure 4: Firms that contribute the most to textual volume.

## 3.2 Exploration of Gender and Pronouns

Numerous studies have shown that NLP systems tend to propagate gender bias found in their training corpora [9, 11, 7]. Along these lines, we explore gender biases in BeanCounter by computing the percentage of filings which contain at least one instance of a specific type of pronoun. Table 2 presents results. Similar to prior work, we observe that *He* pronouns are over-represented relative to *She* pronouns [13, 55]. This could mean that models trained on BeanCounter may generate *He* pronouns at a higher rate than *She* pronouns. We also observe that 2nd person pronouns are used roughly 20% less frequently–on an absolute basis–than in the dataset of Touvron et al. [55].

Table 2: Percentage of filings in BeanCounter that contain gender and grammatical pronouns. 68.2% of all filings contain gender pronouns, and 29.8% of this subset contains *She* pronouns.

| Gender Pronouns | 68.23% | Grammatical Pronouns | 94.49% |
|---|---|---|---|
| **She** (she, her, hers, herself) | 29.81% | **1st Person** (I, me, my, mine, myself, we ...) | 69.22% |
| **He** (he, him, his, himself) | 48.51% | **2nd Person** (you, your, ...) | 43.78% |
| **Unknown** (they, them, their ...) | 93.76% | **3rd Person** (it, its, itself, she ...) | 95.57% |

## 3.3 Exploration of Demographic Identities

Similar to Touvron et al. [55], we measure demographic representation in BeanCounter by aggregating observations of descriptors from the HolisticBias dataset [50]. Specifically, we compute the number of filings that contain at least one specific demographic descriptor across five axes: Gender and Sex, Sexual Orientation, Nationality, Race and Ethnicity, and Religion. We remove several demographic terms such as "straight", "white", "Black", "bi", "pan", "ace" and "poly" because these terms can have multiple meanings and after manual inspection we found these are generally used in ways other than as demographic descriptors. We also include expanded versions of abbreviated descriptors (e.g., "AFAB" and "Assigned-Female-At-Birth") as well as terms that are often hyphenated or capitalized (e.g. "African-American" and "African American"; "Male-to-Female" and "male-to-female") to capture as many mentions of these descriptors as possible. We note that while most of the terms are used in the intended context, some terms may be used in contexts outside of demographic identity, e.g., "Christian" is also a common English name, and we do not exclude these from the analysis.

In comparison to the pretraining corpuses of Touvron et al. [55] and Dodge et al. [16], Table 3a shows that BeanCounter has significantly more representation of content along the *Nationality* and *Race and Ethnicity* axes, whereas it has a significantly lower representation under the *Sexual Orientation* axis.[3] Similar to the two aforementioned datasets, we observe that BeanCounter is also skewed towards over-representation of Western identities such as "American", "European" and "Christian."

## 3.4 Toxicity Analysis of Content Associated with Demographic Identities

To our knowledge, toxicity has not been explored in business-oriented corpuses. We extend this literature by exploring toxic language surrounding the demographic descriptors identified in the previous prevalence analysis (Section 3.3). Prior work on toxicity analysis typically focuses on assigning a toxicity score to randomly sampled spans of 100 tokens [45] or computing document-level average toxicity by scoring each line of a document separately [55]. Our approach improves upon previous approaches in several ways.

First, prior literature has identified that toxic language is relatively rare, e.g., 0.2% of documents in the corpus of Touvron et al. [55] are labeled as toxic. Thus, randomly sampling text spans could omit highly toxic samples and hence underestimate the toxicity of the dataset. Further, even though highly toxic text may be rare, it does not mean such text is harmless or has limited downstream impact. Rather than randomly sampling parts of BeanCounter to measure toxicity, *we focus on instances where toxicity is likely to occur*: around demographic descriptors[25, 50]. As a result, one can view our analysis as providing evidence on the question "conditional on the existence of a demographic descriptor, how likely is the surrounding content to be toxic?"

Second, most SOTA toxicity classifiers are trained on comments from various web-based sources [3, 12] or machine generated sentences about minority groups [25]. As a result, they are best equipped to evaluate toxicity on texts of sentence lengths and are likely to make erroneous predictions for larger text spans such as those from breaking a document by new-line characters. We address this issue by identifying sentences containing these descriptors and passing them into the Perspective API [3]. Manual inspection of classification results suggests that this leads to more accurate results relative to attempting to classify longer snippets or using other models such as ToxiGen-HateBERT [25].

---

[3]Touvron et al. [55] perform a similar analysis in their demographic representations analysis. To compare BeanCounter's demographic composition to these datasets, we perform the same analysis on C4-en, and its demographic composition is availabe in Appendix C.

Table 3: Demographic representation and toxicity

| Axes | Descriptor | % Filing |
|---|---|---|
| Gender and Sex (9.84%) | feminine | 67.74% |
| | masculine | 67.51% |
| | female | 25.01% |
| | male | 21.33% |
| | gender neutral | 1.55% |
| Sexual Orientation (0.32%) | LGBTQ | 55.69% |
| | gay | 32.66% |
| | lesbian | 28.28% |
| | LGBT | 16.12% |
| | bisexual | 15.85% |
| Nationality (79.29%) | American | 91.58% |
| | Chinese | 11.28% |
| | Mexican | 6.51% |
| | Indian | 6.13% |
| | Korean | 3.27% |
| Race and Ethnicity (49.96%) | European | 81.16% |
| | Latin | 18.93% |
| | Asian | 18.70% |
| | African | 6.60% |
| | Latin American | 5.95% |
| Religion (9.52%) | religious | 30.19% |
| | Christian | 29.36% |
| | secular | 16.00% |
| | Catholic | 13.82% |
| | Methodist | 11.97% |

| Axes | Descriptor | % Reduction |
|---|---|---|
| Gender and Sex | masculine | -63.85% |
| | female | -67.70% |
| | male | -72.61% |
| | feminine | -66.45% |
| | gender neutral | -66.01% |
| Sexual Orientation | LGBTQ | -85.68% |
| | LGBT | -84.82% |
| | gay | -65.79% |
| | lesbian | -89.74% |
| | heterosexual | -60.79% |
| Nationality | American | -80.38% |
| | Chinese | -72.33% |
| | Mexican | -81.25% |
| | Indian | -75.91% |
| | Korean | -78.01% |
| Race and Ethnicity | European | -59.11% |
| | Latin | -76.76% |
| | Asian | -77.03% |
| | African | -74.63% |
| | Arab | -80.16% |
| Religion | Christian | -87.48% |
| | secular | -82.27% |
| | Catholic | -85.69% |
| | religious | -66.53% |
| | Methodist | -74.61% |

(a) The percentage listed under each demographic axis is the percentage of filings that contain a demographic descriptor under the particular axis. The percentage next to each demographic descriptor represents the percentage of filings that has at least one mention of the particular descriptor, given that it mentions a descriptor of the particular axis. See Table 6 for the most prevalent descriptors in C4-en.

(b) For each axis, we compute the average toxicity scores of the sentences containing the 5 most frequently mentioned descriptor in BeanCounter and C4-en. The average toxicity scores of these sentences in Bean-Counter is 59-89% lower than those in C4-en. All % reduction are statistically significant at the 1%-level (two tailed) except for 'heterosexual'. For average toxicity scores of BeanCounter and C4-en, see Table 7.

Specifically, for every sentence in BeanCounter, we use a regex statement to determine whether the sentence contains any demographic descriptors; if there are multiple descriptors in a sentence, the sentence is assigned to the longest descriptor, i.e., if a sentence contains the descriptor "Latin American", it would be assigned to "Latin American" instead of "Latin" or "American". This is a generally reliable heuristic since most of the extracted sentences contain one descriptor; however, when sentences contain multiple distinct descriptors, it becomes unclear which descriptor is the primary "target." We follow the same procedure to extract all sentences with descriptors from the C4-en dataset. Since C4-en has a different frequency of descriptors than BeanCounter, we generate a balanced sample from C4-en which matches the frequency in BeanCounter. In particular, for each descriptor $d_i$, we count the number of sentences $n_i$ in BeanCounter which contain $d_i$. We then randomly sample $n_i$ sentences from C4-en which contain $d_i$. We find that the average sentence length is 54 tokens, which is of comparable length to the training data of toxicity classifiers [3, 12].

Next, we assign each sentence a toxicity score between 0 and 1 using the Perspective API [3]. Table 3b presents the difference in average toxicity between C4-en and BeanCounter on the most prevalent descriptors in each demographic axis. We find that the average toxicity of sentences containing a given descriptor in BeanCounter is at least 59% lower than those from C4-en. In addition to quantitatively exploring differences in toxicity, we also read and compare examples of toxic content

from C4-en and BeanCounter. We find that toxic content in BeanCounter tends to come from the discussion of business models which involve breeding animals, medical ailments which affect certain portions of the population at different rates, firms providing examples of actions that would violate their ethics policies, and entities raising money to produce certain artistic works, e.g., movies, which contain content labeled as toxic. In contrast, the toxic content in C4-en tends to be directed at specific groups. Appendix E presents a sample of the most toxic examples from both datasets.

Considering other datasets, Rae et al. [45] also uses Perspective API and reports mean and median toxicity scores of 0.10 and 0.07, respectively, for their random sample of text spans. Even when explicitly sampling text likely to contain toxic language, toxicity of BeanCounter is an order of magnitude lower: the mean and median toxicity score of BeanCounter's descriptor sentences are 0.009 and 0.006, respectively. This suggests that BeanCounter can serve as a large-scale corpus of very low toxicity text.

## 4 Evaluation

Next, we explore the potential utility of BeanCounter along two dimensions: (i) domain-specific applications within finance and (ii) generation toxicity. To facilitate this analysis, we continue pre-training Pythia-1.4B [8] and Phi-1.5 [36] for 1B tokens sampled from BeanCounter stratified by year. We choose these models because Pythia-1.4B is a well-studied model which is known to generate toxic content and Phi-1.5, while newer, has demonstrated excellent performance across a number of domains and exhibits comparatively low toxicity. We use the same optimization parameters used for initial pre-training of these models except that we reduce the maximum learning rate by 50% and cosine decay to 10% of the maximum without any warm-up phase [24]. Details of financial, toxicity and general LLM evaluation benchmarks can be found in Appendix B.

### 4.1 Financial Domain

To understand whether BeanCounter can improve a model's finance domain knowledge, we evaluated the models on two widely used benchmarks: Financial Phrasebank (FPB) [40] and Fin NER[6]. FPB is a sentiment classification task on 4840 sentences sampled from financial news[40]. The sentences are assigned "positive", "neutral" and "negative" labels according to how it will impact the mentioned company's stock prices. Fin NER is a named entity recognition task consisting of 1467 labeled sentences extracted from financial agreements filed on the SEC [6].

As seen in Table 4, we find larger improvements in the Phi-1.5 model after continued pretraining on BeanCounter; there is a 4.3% improvement on the Fin NER task and 2.5% improvement on FPB. The improvements for Pythia-1.4B are 1.4% and 1.1% for Fin NER and FPB respectively.

Table 4: Model performance on financial domain and toxicity tasks

|  | Fin NER | FPB | RealToxicityPrompts | |
|---|---|---|---|---|
|  | F-1 score | F-1 score | Toxicity score | Perspective score |
| Pythia-1.4B | 0.72 | 0.92 | 0.0579 | 0.1297 |
| Pythia-1.4B-BeanCounter | **0.73** | **0.93** | **0.0385** | **0.1118** |
| Phi-1.5 | 0.71 | 0.93 | 0.0224 | 0.0901 |
| Phi-1.5-BeanCounter | **0.74** | **0.95** | **0.0184** | **0.0887** |

### 4.2 Generation Toxicity

To quantify the models' propensity for toxic generation, we evaluate them on RealToxicityPrompts [20] and SafeNLP [28]. RealToxicityPrompts is a set of 100K prompts extracted from sentences in the OpenWebText corpus intended to illicit toxic generation [20]. The generation is assigned a toxicity score of 0 if the Perspective API [3] score is < 0.5 and 1 otherwise. SafeNLP uses a subset of the ToxiGen dataset [25] to compute safety scores across 13 marginalized demographics for a pre-trained language model [28].

Table 4 presents results. We find a reduction of 18-33% in Toxicity score after continued pretraining on BeanCounter. In terms of safety scores in Figure 5, these increase by 6-21% across demographics for Pythia-1.4B and by 3-7% for Phi-1.5 except for "jewish," "latino," "mexican," and "physical disability." It is important to note that although higher safety scores indicate less propensity to generate toxic content, it does not mean that the models are immune to harmful generations.

We hypothesize that the reduction in toxic generation results from (i) multi-turn conversations between firms and shareholders discussing topics associated with toxicity and/or (ii) firms presenting and explaining examples of violations of their ethics policies. For example, Section E.1 presents a discussion between the CEO of PepsiCo and a shareholder. While the discussion is some of the more toxic content in BeanCounter, it is still comparatively civil and each side explains their point of view. If LLMs can learn from such multi-turn conversations then this may be one reason we observe an improvement in safety scores.

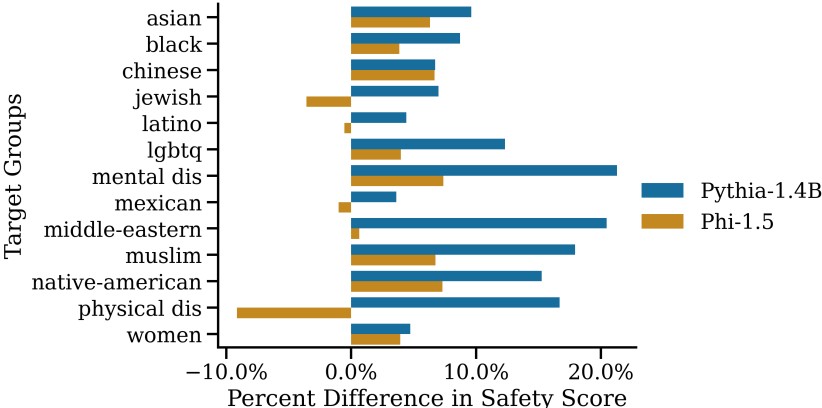

Figure 5: Changes in safety scores across 13 demographic target groups for Pythia-1.4B and Phi-1.5 after continued pre-training on BeanCounter. The safety score ranges from 0 to 1 where a higher score indicates a lower likelihood for the model to produce toxic generation relative to benign generation.

# 5 Limitations

**Ablation**   We evaluated the impact of continued pre-training with BeanCounter on two small LLM models (1.3B and 1.4B parameters for Phi-1.5 and Pythia-1.4B respectively). It is unclear if Bean-Counter would have the same, lesser, or greater impact on model performance with different baseline model architectures, larger sizes, or training data mixes, e.g., pre-training solely on BeanCounter. Future work could explore additional models and incorporating BeanCounter into the training data mix at various percentages.

**Deceptive Content**   The content in BeanCounter is produced by businesses with economic incentives to communicate a specific image of their business within the confines of the regulatory environment. As a result, the content may comply with regulations while being subtly misleading. Training on such data could lead to models which are misaligned with users' preferences, e.g., models which intentionally deceive the user in ways that are difficult to detect. In more extreme cases, the content may be entirely false, e.g., in the case of frauds such as ENRON and Wirecard. While this type of deceptive content is likely present in web-based datasets, researchers should be aware that the more regulated environment in which the contents of BeanCounter was produced does not guarantee that the data is free of deceptive content. The "BeanCounter.fraud" split of the data containing content from known frauds could enable future research to detect deceptive content.

**Generalizability**   While this work suggests that BeanCounter is a novel source of low-toxicity data, the data is sourced from an environment that differs substantially in content and style from the web. Along these lines, we find that some descriptors have different meanings within the business domain, and the prevalence of certain demographics differs from prior datasets. As a result, models trained solely on BeanCounter may generate text which lacks imagination, is not desirable for a broad

audience, fails to capture the meaning of certain words, and/or exhibits biases. Related to biases, there are limitations to our exploration of toxicity and demographic identities. Specifically, we only explored a subset of demographic identity descriptors from five axes. Future works could explore the additional axes, e.g., Ability, Socioeconomic Status and Age, to understand how these identities are represented in NLP datasets.

**Out-of-Domain Performance** While our work suggests that continued pre-training on BeanCounter can improve performance on financial related tasks, this may come at the expense of reduced performance on tasks unrelated to BeanCounter's content. Results for the two small LLMs we consider are mixed. Table 5 illustrates the models' performance on a suite of common general LLM comprehension and reasoning tasks. Pythia-1.4B models seem to perform similarly or slightly better on these benchmarks after continued pre-training, whereas Phi-1.5 models seem to experience decreases in performance in some tasks. Hence, the impact of continued pre-training on BeanCounter may be model-specific, or there may be interactions between the datasets used to train the models (e.g. Phi-1.5 is trained on a novel mix of data [36]).

## 6    Conclusion

In this work, we present BeanCounter, a 159B token dataset of business-oriented text extracted from publicly available financial disclosures. To our knowledge, BeanCounter is the largest public corpus in the business domain. Relative to datasets derived from the web, the content in BeanCounter is more likely to be truthful and of high quality because the entities producing the content face civil and criminal penalties if it is not. Additionally, each piece of content in BeanCounter is associated with a timestamp representing when the content was made available thereby facilitating future research on the role of time in language models. Lastly, we explore biases and toxicity in the data using a novel evaluation scheme which focuses on locations where biased and/or toxic content is most likely to be present. We find that BeanCounter is significantly less toxic compared to another widely used corpus, and our preliminary exploration of using BeanCounter for continued pre-training shows improvements in finance domain knowledge and reduced toxic generation.

## Acknowledgments and Disclosure of Funding

We acknowledge generous financial support from the Booth School of Business and the Center for Applied AI. The authors have no competing interests to disclose.

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

# A    Additional Details of Dataset Construction

While our approach generally follows the best practices for data curation, filtering, and deduplication which have been developed in recent years, this appendix provides details specific to the construction of BeanCounter. An overview of the four high-level steps involved in constructing BeanCounter is presented in Figure 1 and each step is detailed below. Figure 6 presents an example disclosure which highlights our approach to constructing BeanCounter relative to the source document and prior literature.

*Item 1. Business (Continued)*

Our wholesale unit volumes vary with the level of total industry demand and our share of that industry demand. Our wholesale unit volumes also are influenced by the level of dealer inventory. Our share is influenced by how our products are perceived in comparison to those offered by other manufacturers based on many factors, including price, quality, styling, reliability, safety, fuel efficiency, functionality, and reputation. Our share also is affected by the timing and frequency of new model introductions. Our ability to satisfy changing consumer preferences with respect to type or size of vehicle, as well as design and performance characteristics, affects our sales and earnings significantly.

As with other manufacturers, the profitability of our business is affected by many factors, including:

- Wholesale unit volumes
- Margin of profit on each vehicle sold - which in turn is affected by many factors, such as:
  - Market factors - volume and mix of vehicles and options sold, and net pricing (reflecting, among other factors, incentive programs)
  - Costs of components and raw materials necessary for production of vehicles
  - Costs for customer warranty claims and additional service actions
  - Costs for safety, emissions, and fuel economy technology and equipment
- A high proportion of relatively fixed structural costs, so that small changes in wholesale unit volumes can significantly affect overall profitability

(a) HTML rendering of filing on EDGAR

```
Item 1. Business (Continued)
Our wholesale unit volumes vary with the level of total industry demand and our share of that industry demand. Our wholesale unit volumes also are
influenced by the level of dealer inventory. Our share is influenced by how our products are perceived in comparison to those offered by other
manufacturers based on many factors, including price, quality, styling, reliability, safety, fuel efficiency, functionality, and reputation. Our
share also is affected by the timing and frequency of new model introductions. Our ability to satisfy changing consumer preferences with respect to
type or size of vehicle, as well as design and performance characteristics, affects our sales and earnings significantly.
As with other manufacturers, the profitability of our business is affected by many factors, including:
•
Wholesale unit volumes
•
Margin of profit on each vehicle sold — which in turn is affected by many factors, such as:
◦
Market factors — volume and mix of vehicles and options sold, and net pricing (reflecting, among other factors, incentive programs)
◦
Costs of components and raw materials necessary for production of vehicles
◦
Costs for customer warranty claims and additional service actions
◦
Costs for safety, emissions, and fuel economy technology and equipment
•
A high proportion of relatively fixed structural costs, so that small changes in wholesale unit volumes can significantly affect overall
profitability
```

(b) Text extracted by EDGAR-CORPUS

```
    Our wholesale unit volumes vary with the level of total industry demand and our share of that industry demand.  Our wholesale unit volumes also
are influenced by the level of dealer inventory.  Our share is influenced by how our products are perceived in comparison to those offered by other
manufacturers based on many factors, including price, quality, styling, reliability, safety, fuel efficiency, functionality, and reputation.  Our
share also is affected by the timing and frequency of new model introductions.  Our ability to satisfy changing consumer preferences with respect to
type or size of vehicle, as well as design and performance characteristics, affects our sales and earnings significantly.

    As with other manufacturers, the profitability of our business is affected by many factors, including:

    •Wholesale unit volumes
    •Margin of profit on each vehicle sold — which in turn is affected by many factors, such as:
        ◦    Market factors — volume and mix of vehicles and options sold, and net pricing (reflecting, among other factors, incentive programs)
        ◦    Costs of components and raw materials necessary for production of vehicles
        ◦    Costs for customer warranty claims and additional service actions
        ◦    Costs for safety, emissions, and fuel economy technology and equipment
    •A high proportion of relatively fixed structural costs, so that small changes in wholesale unit volumes can significantly affect overall
profitability
```

(c) Text extracted by BeanCounter

Figure 6: Text extracted from the same section in Ford's 2017 10-K filing (filed 2018-02-08 ToDo: Check date). The extraction scheme in BeanCounter better preserves indentation and and list format in comparison to EDGAR-CORPUS. "Item 1. Business (Continued)" is also excluded from the text in BeanCounter since the extraction scheme removes repetitious text at the start and end of each page.

## A.1    Collection

The pipeline begins with downloading daily archives of all submissions to EDGAR. These archives contain the full contents of each submission, i.e., the main filing as well as any and all attachments, and are produced daily by the SEC. Availability of archives begins on 1996-01-12 and continues through to the present, although BeanCounter's coverage is only through the end of 2023. The submissions in these archives also contain metadata on the submission such as the time the submission was accepted by EDGAR. This metadata is parsed and included in BeanCounter. In total, these archive files contain 3TB (compressed) of content.

## A.2 Extraction

After downloading all filings, the pipeline extracts text from the main filing and all attachments. While EDGAR supports a variety of attachments such as plain text, HTML, and images, the pipeline only extracts content from text and HTML-based documents since these documents represent the majority of the content on EDGAR. Historically, EDGAR accepted text-based filings which had been formatted in a fixed-width style. This is in contrast to modern HTML files where the browser is largely responsible for formatting the content to match a particular device. For this reason, the pipeline extracts content from text and HTML-based filings using distinct logic.

Text-based filings consist of both text and standard generalized markup language (SGML). These SGML tags are used to denote pages, tables, and indicate other formatting details for older devices. The first step of processing text-based filings is to remove all numeric tables from the filing. SGML tags indicating page breaks are then used to identify sentences spanning multiple pages and "unbreak" them. Lastly, since these filings were prepared for fixed-width screens, e.g., lines were wrapped after 80 characters, we unwrap all lines and replace sequences of more than two newline characters with just two.

HTML-based filings are processed using the Beautiful Soup [2] library. As with text-based filings, the pipeline first removes all numeric tables from the filing. Since table tags are frequently used in HTML to structure content, we cannot rely on the mere presence of a table tag as indicia of a numeric table. Instead, for each table identified, we compute a measure of text-density, characters per tag (CPT), using only alphabetical characters [52]. Any table with a CPT less than 10 is removed. Via manual inspection, we find that this approach is high precision in the sense that there are few false positives.

After removing tables, we extract text from the remaining content while taking care to preserve formatting such as indentation and spacing. Where there are page breaks, we again "unbreak" sentences spanning multiple pages and remove any "header" content. The effects of our approach can be seen in Figure 6: Panel 6a presents the disclosure as rendered by a web browser, Panel 6b presents the extracted text in EDGAR-CORPUS [39], and Panel 6c presents the extracted text in BeanCounter. The first notable difference between our approach and prior literature is that we preserve white-space according to HTML and CSS directives.

The second notable difference is our removal of header content. Note that in Panels 6a and 6b, the text "Item 1. Business (Continued)" appears at the top of the page. Similar text is repeated at the top of each page to indicate the current section. Simply extracting the text without any consideration of this header content would lead to situations where a sentence spanning multiple pages has header content injected mid-sentence. Our approach identifies such content and removes it, as can be seen in Figure 6c.

## A.3 Cleaning

Prior work has used heuristics such as line length and terminal punctuation to identify and remove low-quality content, e.g., Raffel et al. [46]. While there is such content in EDGAR filings, we find that in many instances this content is relevant to the long-form text surrounding such lines which would be removed under these heuristics. For example, many filings contain lists enumerating risks facing the business or on-going activities. The elements of these lists often do not end in punctuation and/or are relatively short–as can be seen in Figure 6a. Since the content on EDGAR is meant to convey important value-relevant information, the entities producing content for EDGAR are more likely to dedicate significant resources to ensuring it is high-quality. Along these lines, we adopt a lighter approach to cleaning than some of the prior literature.

Specifically, we exclude (i) all filings which are of a type known to be standardized forms containing little long-form narrative text, (ii) all documents with fewer than 200 words, and (iii) all documents where the proportion of whitespace is more than the 99-th percentile for the whole dataset (41%). We refer to the resulting dataset as "BeanCounter.clean" and find that it contains more than 159B tokens.

## A.4 Deduplication

Following prior literature, e.g., Lee et al. [32], we deduplicate BeanCounter at the document level using MinHash and LSH. Specifically, we convert each attachment into 5-grams which is then hashed

using 260 permutations (20x13). Near duplicates are identified using an LSH index with a similarity threshold set to 80%. In cases where the set of near duplicates is too large to compute the true duplicates, e.g., a set of a million attachments, we assume that all near duplicates are in fact true duplicates. While this is not accurate, it is conservative in the sense that we remove some attachments which are not actually duplicates. For each set of attachments identified as duplicates, we keep the attachment which was first accepted by EDGAR, i.e., the attachment with the earliest acceptance timestamp. We refer to the resulting dataset as "BeanCounter.final" and find that it contains more than 111B tokens.

## B    Additional Details of Model Evaluation

This section details the various financial, toxicity and general benchmarks used to evaluate the LLMs and their continually pre-trained variants. To ensure replicability of our results, we use libraries and harnesses that are publicly available. For continued pre-training and fine-tuning we use a combination of local and cloud hosted NVIDIA RTX 4090, A100, and H100 GPUs for a total of roughly 400 GPU-hours.

Table 5: Model Performance on General LLM Benchmarks in Normalized Accuracy

|  | BBH | MMLU-Pro | MuSR | IFeval | GPQA |
|---|---|---|---|---|---|
| Pythia-1.4B | **0.3174** | 0.1125 | 0.3498 | **0.2397** | 0.2495 |
| Pythia-1.4B-BeanCounter | 0.3135 | **0.1134** | **0.3645** | 0.2375 | **0.2674** |
| Phi-1.5 | **0.3346** | **0.17** | 0.3404 | **0.2051** | **0.2732** |
| Phi-1.5-BeanCounter | 0.3124 | 0.1252 | **0.3471** | 0.1928 | 0.2479 |

**Financial Phrasebank (FPB)** [40]: This task requires the model to classify the sentiment of text from the financial domain. Specifically, Malo et al. [40] create a dataset consisting of 4844 sentences that mentions a specific company randomly selected from financial news articles on LexisNexis database. The sentences are annotated by 16 annotators with finance-related background (e.g. 3 researchers and 13 Master's students majoring in finance, accounting and economics) and they assigned "positive", "neutral" or "negative" labels according to how they perceive the information in the sentences will impact the mentioned company's stock price. The sentence labels have varying levels of agreement across the annotators, and we evaluated the models on the subset (2264 sentences) of the original dataset with 100% agreement amongst the annotators. We finetuned the model, with an AdamW optimizer with learning rate of 5e-5, for 3 epochs on the training set and used the model with the highest weighted F-1 score for evaluation on the validation set.

**Fin NER** [6]: In this task, the model is suppose to complete NER labeling on sentences extracted from financial agreements. To construct the dataset, Alvarado et al. [6] randomly selected 8 financial agreements filed on the SEC EDGAR for manual annotation, based on the name entity types in the CoNLL-2003 dataset: location (LOC), organization (ORG), person (PER), and miscellaneous (MISC). To evaluate the models, we used the T-NER library [58]; the models are fine-tuned for 10 epochs with grid search on the training and validation set to find the set of hyper-parameters that yields the highest F-1 score. The model with the highest F-1 score is then evaluated on the test split of Fin NER.

**RealToxicityPrompts** [20]: The purpose of this task is to understand how likely it is for the model to produce toxic content when the selected prompts are prone to illicit toxic generation. This benchmark consists of 99k prompts extracted from sentences in the OpenWebText corpus [21], where 22% of the prompts are classified as toxic, i.e. being assigned a toxicity score $\geq 0.5$ by Perspective API [3], and the average toxicity score of the prompts is 0.29. The models are given a prompt and generation is run until a newline character is produced. The generation is assigned a toxicity score of 0 if the Perspective API score is < 0.5 and 1 otherwise. In Table 4, the "Toxicity score" column indicates the percentage of generations that are classified as toxic by the aforementioned threshold, whereas the "Perspective score" column is an average of the raw output score of Perspective API, which ranges from 0 to 1 where 1 is highly toxic. We used the lm-evaluation-harness [19] from EleutherAI to evaluate the models on RealToxicityPrompts.

**SafeNLP** [28]: This evaluation task uses a subset of the ToxiGen dataset [25] to compute safety scores across 13 marginalized demographics for a pre-trained language model. The scaled perplexity of each statement is computed by the model's perplexity divided by the pre-computed toxicity score of the statement. Lastly, the scaled perplexity values of harmful and benign sentences of each target group is fed into the Mann-Whitney U-test to compute a final safety score for that demographic. A non-toxic pre-trained model should have high scaled perplexity for implicitly harmful sentences and low scaled perplexity for benign sentences. We used the evaluation package provided by Hosseini et al. [28] to compute the safety scores for our models.

**Big Bench Hard (BBH)** [53]: This benchmark includes a suite of 23 challenging BIG-Bench tasks where prior LLMs have yet to outperform the average human-rater. Some example tasks include evaluating Boolean expressions, logical deduction of object position and other "riddle-like" tasks. This evaluation and the following evaluations are performed with lm-evaluation-harness [19] from EleatherAI on the leaderboard setting, which is the setting used for evaluations on the Huggingface Open LLM Leaderboard. For ease of comparison and presentation, the accuracy we present in Table 5 is an average across the 23 tasks.

**Massive Multitask Language Understanding (MMLU-Pro)** [60]: This evaluation is an extension of MMLU, a multiple choice benchmark focused on language comprehension and reasoning across varying domains such as Math, Law, Health, Business, etc. Compared to MMLU, the MMLU-Pro benchmark contains more challenging, reasoning-focused questions and includes 10 choices instead of the original 4 choices. We ran the evaluation on a 5-shot setting via lm-evaluation-harness [19], which is the setting used in the Huggingface Open LLM Leaderboard.

**Multistep Soft Reasoning (MuSR)** [51]: This reasoning evaluation focuses on evaluating the model's ability to perform multistep soft reasoning tasks in a natural language narrative context. One of the most challenging subtasks is murder mysteries, which includes an approximately 1,000 word story where the model is prompted to identify the killer. MuSR combines commonsense and multistep reasoning, which is lacking in prior LLM benchmarks. The reported accuracy in Table 5 is an average over 3 sub-tasks: murder mysteries, object placements and team allocation.

**Instruction-Following Eval (IFeval)** [64]: This instruction following benchmark focuses on evaluating models on a set of easily "verifiable instructions" such as "mention the keyword of AI at least 3 times". Each prompt contains one or more verifiable instruction, and the accuracy presented in Table 5 is an average of "prompt-level" and "instruction-level" accuracies.

**Graduate-Level Google-Proof Q&A (GPQA)** [47]: This evaluation task consists of 448 challenging multi-choice questions written by domain-experts who have or are pursuing PhDs in biology, physics and chemistry. For highly skilled human non-experts (i.e. experts in another field but not the tested fields) with access to internet resources, they reach 34% accuracy after spending an average of 37 minutes trying to answer each question. The accuracy reported in Table 5 is an average over the three sets of questions: GPQA main, GPQA extended and GPQA diamond.

## C  Descriptor prevalence of C4 en and average toxicity score of BeanCounter and C4-en

Table 6: Prevalence of descriptors in C4-en

| Demographic Axes | Descriptor | % Doc |
|---|---|---|
| Gender and Sex (12.9%) | female | 58.10% |
| | male | 43.40% |
| | feminine | 10.20% |
| | masculine | 4.40% |
| | transgender | 2.80% |
| Sexual Orientation (2.6%) | gay | 54.80% |
| | LGBT | 19.20% |
| | lesbian | 14.50% |
| | LGBTQ | 14.50% |
| | queer | 12.20% |
| Nationality (51.1%) | American | 64.20% |
| | Indian | 16.40% |
| | Chinese | 16.20% |
| | Mexican | 4.70% |
| | Korean | 4.50% |
| Race and Ethnicity (30.9%) | European | 45.40% |
| | African | 17.80% |
| | Asian | 15.10% |
| | Latin | 10.10% |
| | Arab | 6.30% |
| Religion (25.8%) | Christian | 31.20% |
| | religious | 23.80% |
| | spiritual | 22.60% |
| | Catholic | 13.10% |
| | Jewish | 11.50% |

Table 7: Toxicity Scores (between 0-1) of Top 5 Most Prevalent Descriptors in BeanCounter and C4

| Axes | Descriptor | Average Toxicity BeanCounter | Average Toxicity C4 |
|---|---|---|---|
| Sexual Orientation | LGBTQ | 0.0052 | 0.0361 |
| | LGBT | 0.0059 | 0.0387 |
| | gay | 0.0320 | 0.0936 |
| | lesbian | 0.0111 | 0.1085 |
| | heterosexual | 0.0543 | 0.1386 |
| Religion | Christian | 0.0096 | 0.0765 |
| | secular | 0.0103 | 0.0580 |
| | Catholic | 0.0075 | 0.0525 |
| | religious | 0.0207 | 0.0619 |
| | Methodist | 0.0053 | 0.0209 |
| Race and Ethnicity | European | 0.0108 | 0.0264 |
| | Latin | 0.0075 | 0.0323 |
| | Asian | 0.0122 | 0.0529 |
| | African | 0.0121 | 0.0478 |
| | Arab | 0.0099 | 0.0501 |
| Nationality | American | 0.0076 | 0.0386 |
| | Chinese | 0.0099 | 0.0358 |
| | Mexican | 0.0111 | 0.0592 |
| | Indian | 0.0073 | 0.0304 |
| | Korean | 0.0083 | 0.0379 |
| Gender and Sex | masculine | 0.0263 | 0.0726 |
| | female | 0.0215 | 0.0665 |
| | male | 0.0195 | 0.0712 |
| | feminine | 0.0295 | 0.0881 |
| | gender neutral | 0.0184 | 0.0541 |

# D Examples of Content from Most Fequent Form Types

In this section we present examples of content from the most common form types in BeanCounter along with descriptions of how these forms are typically used.

## D.1 Forms 485APOS and 485BPOS

Forms 485APOS and 485BPOS are frequently used by investment management companies to satisfy their reporting obligations under Rules 485(a) and 485(b). These forms generally contain information such as past performance of the fund, risks, investment strategies used by the fund, and expenses.

```
WHAT IS THE FUND'S INVESTMENT OBJECTIVE?

The Fund seeks to maximize total return, which consists of appreciation on its
    investments and a variable income stream. While there is no assurance that the
    Fund will achieve its investment objective, it endeavors to do so by following
    the strategies and policies described in this prospectus.

The Fund's Board may change this objective or the Fund's principal investment
    strategies without a shareholder vote. The Fund will notify you in writing at
    least sixty (60) days before making any such change. If there is a material
    change to the Fund's objective or principal investment strategies, you should
    consider whether the Fund remains an appropriate investment for you.

WHAT ARE THE FUND'S PRINCIPAL INVESTMENT STRATEGIES?

To achieve its objective, the Fund will invest at least 80% of its assets in (i)
    securities of U.S. and non-U.S. companies listed on a national securities
    exchange, or foreign equivalent, who invest at least 50% of their underlying
    assets in five or more privately held companies or five or more companies whose
     business is to invest in and lend capital to privately held companies ("Listed
     Private Equity Companies") and (ii) derivatives that otherwise have the
    economic characteristics of Listed Private Equity Companies.

Listed Private Equity Companies may include, among others, business development
    companies, investment holding companies, publicly traded limited partnership
    interests (common units), publicly traded venture capital funds, publicly
    traded venture capital trusts, publicly traded private equity funds, publicly
    traded private equity investment trusts, publicly traded closed-end funds,
    publicly traded financial institutions that lend to or invest in privately held
     companies and any other publicly traded vehicle whose purpose is to invest in
    privately held companies.
```

Figure 7: Excerpt from a Form 485BPOS filed by Financial Investors Trust on 2007-11-20 (SEC Accession No. 0001104659-07-084426)

## D.2 Form 8-K

Form 8-K, or the "Current Report," is frequently used by businesses to report certain types of events in a timely manner. Examples of events which trigger required filing of Form 8-K include reporting of financial results, delisting of a business's stock, departure or appointment of principal officers, and, more recently, cybersecurity incidents.

```
Tarrytown, NY, November 7, 2014 - Progenics Pharmaceuticals, Inc. (NASDAQ:PGNX)
    today announced its results of operations for the quarter and nine months ended
     September 30, 2014.

The Company also announced that following interactions with the FDA, Progenics'
    worldwide collaboration partner, Salix Pharmaceuticals, Ltd., expects to file
    the oral RELISTOR New Drug Application (NDA) by the end of the second quarter
    2015.

Net income for the quarter was $37.0 million or $0.51 diluted per share, compared to
     net loss of $10.5 million or $0.17 diluted per share in the 2013 period. Net
    income for the current nine months was $16.6 million or $0.24 diluted per share,
     compared to net loss of $34.0 million or $0.63 diluted per share in 2013.

Progenics ended the quarter with cash, cash equivalents and securities of $87.44
    million, reflecting a decrease of $0.12 million in the quarter and an increase
    of $19.37 million from 2013 year-end, resulting primarily from a first quarter
    financing and receipt of $7.25 million upon settlement of an arbitration with
    its former RELISTOR licensee for Japan. Quarter-end cash does not include a $40
    .0 million milestone payment for the September 29th FDA approval of RELISTOR
    Subcutaneous Injection for the treatment of opioid induced constipation in
    patients with non-cancer pain, which was received in October.

"Recent months have brought extraordinary progress for RELISTOR, first with the FDA
    approval of subcutaneous RELISTOR and now with the plan to submit the NDA for
    oral RELISTOR," said Mark Baker, CEO of Progenics. "The approval of
    subcutaneous RELISTOR in chronic pain dramatically increased the number of
    people who can benefit from this remarkable medication and set the stage for
    increased royalty and milestone revenue for Progenics. Now, we look forward to
    bringing these patients a new, more convenient oral formulation. I believe
    these developments will have the potential to transform the treatment of opioid-
    induced constipation."

Third quarter revenue totaled $41.7 million, up from $0.9 million in 2013,
    reflecting an increase in collaboration revenue of $39.9 million, primarily
    resulting from the recognition of the $40.0 million milestone from Salix and a
    $0.9 million increase in RELISTOR royalty income to $1.6 million, compared to
    $0.7 million in the 2013 period. The $7.25 million arbitration settlement is
    reported as third quarter other operating income.
```

Figure 8: Excerpt from a Form 8-K filed by Progenics Pharmaceuticals on 2014-11-07 (SEC Accession No. 0000835887-14-000107)

## D.3 Form 10-Q

Form 10-Q, or the "Quarterly Report," is frequently used by businesses to provide information on a quarterly basis in between the filing of annual reports. These forms generally contain information such as financial statements, discussion of financial condition by the management, legal proceedings, and risks.

```
Management concluded that there is substantial doubt about our ability to continue
    as a going concern. This doubt about our ability to continue as a going concern
     for at least twelve months from the date of issuance of the financial
    statements could materially limit our ability to raise additional funds through
     the issuance of new debt or equity securities or otherwise. Future reports by
    our independent registered accounting firm on our financial statements may also
     include an explanatory paragraph with respect to our ability to continue as a
    going concern. We have incurred significant losses since our inception and have
     never been profitable, and it is possible we will never achieve profitability.
     We believe, based on our current operating plan, that our cash and cash
    equivalents as of September 30, 2022 of approximately $12.4 million, as well as
     future cash flows from net sales of Gimoti, will be sufficient to fund our
    operations into the second quarter of 2023. This period could be shortened if
    there are any significant increases in planned spending other than anticipated.
     We anticipate we will be required to raise additional funds in order to
    continue as a going concern. Because our business is entirely dependent on the
    success of Gimoti, if we are unable to secure additional financing or identify
    and execute on other development or strategic alternatives for Gimoti or our
    company, we will be required to curtail all of our activities and may be
    required to liquidate, dissolve or otherwise wind down our operations. Any of
    these events could result in a complete loss of your investment in our
    securities.

These estimates of cash runway could be shortened if there are any significant
    increases in planned spending on commercialization activities, including for
    marketing and manufacturing of Gimoti, and our selling, general and
    administrative costs to support operations. There is no assurance that other
    financing will be available when needed to allow us to continue as a going
    concern. The perception that we may not be able to continue as a going concern
    may cause others to choose not to deal with us due to concerns about our
    ability to meet our contractual obligations.

On December 29, 2021, we received a letter from Nasdaq indicating that, for the last
     thirty consecutive business days, the bid price for our common stock had
    closed below the minimum $1.00 per share requirement for continued listing on
    the Nasdaq Capital Market.

In accordance with Nasdaq listing rules, we were provided an initial period of 180
    calendar days, or until June 27, 2022, to regain compliance. The letter states
    that Nasdaq will provide written notification that we have achieved compliance
    with its rules if at any time before June 27, 2022, the bid price of our common
     stock closes at $1.00 per share or more for a minimum of ten consecutive
    business days. The Nasdaq letter had no immediate effect on the listing or
    trading of our common stock and the common stock continued to trade on The
    Nasdaq Capital Market.

On April 27, 2022, our stockholders granted the board of directors the authority to
    effect a reverse stock split of our outstanding common stock. On May 23, 2022,
    we effected a 1-for-12 reverse stock split of the shares of our common stock,
    or the Reverse Stock Split. [...]
```

Figure 9: Excerpt from a Form 10-Q filed by Evoke Pharma Inc on 2022-11-09 (SEC Accession No. 0000950170-22-023912)

## D.4  Form 10-K

Form 10-K, or the "Annual Report," is frequently used by businesses to provide information on a annual basis.  These forms generally contain similar information to that in quarterly reports (see Section D.3) while also providing additional details such as a detailed description of the business, its properties, and executive compensation.

```
Overview

    We are a biopharmaceutical company focused on developing and commercializing
        transformative treatments for people affected by rare and debilitating
        diseases.

    Our product, PROCYSBI (cysteamine bitartrate) delayed-release capsules, or
        PROCYSBI, received marketing approval from the U.S. Food and Drug
        Administration, or FDA, in April 2013 for the management of nephropathic
        cystinosis in adults and children six years and older. In Europe, PROCYSBI
        gastro-resistant hard capsules of cysteamine (as mercaptamine bitartrate),
        received a marketing authorization in September 2013 from the European
        Commission, or EC, as an orphan medicinal product for the management of
        proven nephropathic cystinosis in the European Union, or EU. The EU
        marketing authorization allows us to commercialize PROCYSBI in the 28 Member
         States of the EU plus Norway, Liechtenstein and Iceland (which are not EU
        Member States but are part of the European Economic Area, or EEA). PROCYSBI
        received seven and ten years of market exclusivity due to its designation as
         an orphan drug in the United States and the EU, respectively. We achieved
        first commercial sales of PROCYSBI in the United States in June 2013 and in
        the EU, specifically in Germany, in April 2014.

    As of December 31, 2014, insurers of U.S. commercial patients reimburse Raptor
        for PROCYSBI therapy at a Wholesale Acquisition Cost, or WAC, price for
        PROCYSBI of $16,650 per bottle of 250 75-mg capsules and $3,996 per bottle
        of 60 25-mg capsules. Prices for PROCYSBI therapy vary among patients
        because doses are individually based on a patient's weight. In September
        2013, we executed an agreement to participate in the U.S. State Medicare/
        Medicaid rebate program, which will be reflected in our net revenues in
        mandatory rebates on reimbursements for patients receiving state Medicare
        and Medicaid insurance coverage. As of December 31, 2014, our price to
        German, Swiss and Austrian pharmacies was 5,850.23 per bottle of 250 75-mg
        capsules and 468.02 per bottle of 60 25-mg capsules.

Cysteamine Mechanism of Action

    Cysteamine, or 2-aminoethanethiol, the active pharmaceutical ingredient in
        PROCYSBI, is a molecule generated naturally in human cells during the
        metabolism of cysteine. Cysteamine is used to construct the key enzymatic
        cofactor involved in energy produced from sugars and lipids. Cysteamine's
        uniquely reactive properties result in many physiological effects when given
         in pharmaceutical doses.
```

Figure 10: Excerpt from a Form 10-K filed by Raptor Pharmaceutical Corp. on 2015-03-02 (SEC Accession No. 0001140361-15-009703)

## D.5 Form 424B2

Form 424B2 is frequently used by businesses when issuing new securities, e.g., stock. These forms generally contain information such as the offering price, details of the securities themselves, and how they will be distributed.

```
Investment Description

UBS AG Trigger Phoenix Autocallable Optimization Securities (the "Securities") are
    unsubordinated, unsecured debt obligations issued by UBS AG ("UBS") linked to
    the common stock or American depository share of a specific company (the "
    underlying stock"). The applicable terms of an offering of the Securities will
    be specified in the relevant final terms supplement you will receive from your
    financial advisor. The general terms are as follows:

-       Unless otherwise specified in the relevant final terms supplement, the
    principal amount of each Security will equal $10.00.
-       UBS will pay a periodic contingent coupon payment if the closing price of
    the underlying stock on the applicable observation date is equal to or greater
    than the coupon barrier. Otherwise, no coupon will be paid for the relevant
    period. The observation dates, contingent coupon rate, coupon barrier and the
    coupon payment dates will be specified in the relevant final terms supplement
    for your Securities.
-       UBS will automatically call the Securities if the closing price of the
    underlying stock on any observation date is equal to or greater than the
    initial price. If the Securities are called, UBS will pay you the principal
    amount of your Securities plus the contingent coupon for the relevant period
    and no further amounts will be owed to you under the Securities.
-       If the Securities are not called prior to maturity, UBS will either pay the
    principal amount of your Securities plus the contingent coupon for the final
    period or, if the closing price of the underlying stock on the final valuation
    date is below the specified trigger price, you will be fully exposed to the
    negative underlying return and UBS will pay you significantly less than the
    full principal amount, if anything, resulting in a loss on your investment that
     is proportionate to the decline of the underlying stock over the term of the
    Securities. The trigger price will be specified in the relevant final terms
    supplement for your Securities and, unless otherwise specified in the relevant
    final terms supplement, the trigger price will be set equal to the coupon
    barrier.

Investing in the Securities involves significant risks. You may lose some or all of
    your principal amount. The contingent repayment of principal only applies if
    you hold the Securities until maturity. Any payment on the Securities,
    including any repayment of principal, is subject to the creditworthiness of UBS.
     If UBS were to default on its payment obligations, you may not receive any
    amounts owed to you under the Securities and you could lose your entire
    investment.

Features
-       Contingent Coupon - UBS will pay a periodic contingent coupon payment if the
     closing price of the underlying stock on the applicable observation date is
    equal to or greater than the coupon barrier. Otherwise, no coupon will be paid
    for the relevant period.
-       Automatically Callable - UBS will automatically call the Securities and pay
    you the principal amount of your Securities plus the contingent coupon
    otherwise due for the relevant period if the closing price of the underlying
    stock on any observation date is greater than or equal to the initial price. If
     the Securities are not called, investors will have the potential for downside
    equity market risk at maturity.
```

Figure 11: Excerpt from a Form 424B2 filed by UBS AG on 2011-06-15 (SEC Accession No. 0001193125-11-165650)

## D.6 Form 497

Form 497 is frequently used by investment management companies to file any piece of information deemed material to an investor or an investment decision. These forms generally contain information such as a fund's investment strategy, risks, service providers, and fees.

```
Investment Adviser
Goldman Sachs Asset Management, L.P. 200 West Street New York, New York 10282

GSAM has been registered as an investment adviser with the SEC since 1990 and is a
    wholly-owned subsidiary of The Goldman Sachs Group, Inc. and an affiliate of
    Goldman, Sachs & Co. ("Goldman Sachs"). Founded in 1869, Goldman Sachs Group,
    Inc. is a publicly-held financial holding company and a leading global
    investment banking, securities and investment management firm. As of June 30,
    2014, GSAM, including its investment advisory affiliates, had assets under
    supervision of approximately $992.5 billion.

The Investment Adviser provides day-to-day advice regarding the Fund's portfolio
    transactions. The Investment Adviser makes the investment decisions for the
    Fund and places purchase and sale orders for the Fund's portfolio transactions
    in U.S. and foreign markets. As permitted by applicable law, these orders may
    be directed to any executing brokers, dealers, futures commission merchants or
    clearing brokers, including Goldman Sachs and its affiliates. While the
    Investment Adviser is ultimately responsible for the management of the Fund, it
     is able to draw upon the research and expertise of its asset management
    affiliates for portfolio decisions and management with respect to certain
    portfolio securities. In addition, the Investment Adviser has access to the
    research and certain proprietary technical models developed by Goldman Sachs (
    subject to legal, internal, regulatory and chinese wall restrictions), and will
     apply quantitative and qualitative analysis in determining the appropriate
    allocations among categories of issuers and types of securities.

The Investment Adviser also performs the following additional services for the Fund:

-       Supervises all non-advisory operations of the Fund
-       Provides personnel to perform necessary executive, administrative and
    clerical services to the Fund
-       Arranges for the preparation of all required tax returns, reports to
    shareholders, prospectuses and SAIs and other reports filed with the SEC and
    other regulatory authorities
-       Maintains the records of the Fund
-       Provides office space and all necessary office equipment and services

MANAGEMENT FEE AND OTHER EXPENSES

As compensation for its services and its assumption of certain expenses, the
    Investment Adviser is entitled to a fee, computed daily and payable monthly, at
     an annual rate listed below (as a percentage of the Fund's average daily net
    assets):
```

Figure 12: Excerpt from a Form 497 filed by Goldman Sachs Variable Insurance Trust on 2014-10-20 (SEC Accession No. 0001193125-14-375909)

## D.7 Form DEF 14A

Form DEF 14A, or the "Proxy Statement," is frequently used by businesses to notify shareholders of matters to be brought to a vote at a shareholders meeting. These forms generally contain details of the business's corporate governance such as votes on hiring of directors and executive compensation.

APPROVAL OF THE AMENDMENT TO THE 2003 NON-EMPLOYEE DIRECTOR STOCK OPTION PLAN

Effective January 2003, our board of directors adopted the Conn's, Inc. 2003 Non-Employee Directors Stock Option Plan. The plan was approved by our stockholders at the January 17, 2003 special meeting of the stockholders of Conn Appliances, Inc., our predecessor corporation. The purpose of the plan is to attract and retain persons of outstanding competence to serve on the board of directors who are not employed by the Company. On March 28, 2006, our board of directors, subject to shareholder approval, amended our 2003 Non-Employee Director Stock Option Plan to increase the number of shares of common stock that may be issued under the plan from 300,000 to 600,000, and to provide for the vesting of the annual option grants after the director's fourth anniversary as a director on the one year anniversary date of the issuance of such options.

General Description of the Plan

Under the plan, non-employee directors are awarded non-qualified stock options as specifically directed and required under the plan. We currently have seven non-employee directors. It is intended that there will be no tax consequences to the non-employee director or the Company by reason of the issuance of the options to the non-employee director. Upon election to the board, each non-employee director is awarded options to acquire 40,000 shares of Company common stock. Each non-employee director is also awarded options to purchase 10,000 shares of our common stock following each annual stockholder meeting after the fourth anniversary of each non-employee director's initial election or appointment to the board. Each option vests in twenty-five percent increments commencing on the first annual anniversary of the grant date, unless otherwise stated in the option agreement granting the options. Currently, each option agreement for all non-employee directors states that the options vest equally over a three year period. If a non-employee director resigns or is not reelected prior to the vesting of all of his/her options, the unvested portion of the options as well as all vested but unexercised within three years of the date of such resignation or non-election is returned to the shares available for issuance.

Options granted under the plan are non-qualified stock options. Subject to early termination provisions, options may have a term of up to 10 years. The exercise price for non-qualified stock options is determined by our board of directors, as the administrator of the plan. Options granted under the plan may not be sold, pledged, assigned, or otherwise disposed of other than by will or by the laws of descent or distribution. During the lifetime of the non-employee director to whom the option was granted, the option may only be exercised by that non-employee director.

The plan, prior to the amendment, provides and the Form S-8 filed with the SEC registered 300,000 shares of Company common stock for issuance under the plan. At January 31, 2006, each of the Company's seven non-employee directors have received the required, under the plan, options to 40,000 shares of company stock, for a total of 258,000 shares of common stock subject to options issued and outstanding under the plan, 22,000 outstanding options had been exercised, and 20,000 shares remain available for issuance subject to options under the plan.

Figure 13: Excerpt from a Form DEF 14A filed by Conn's Inc. on 2006-04-20 (SEC Accession No. 0000950129-06-004138)

### D.8 Form 6-K

Form 6-K is used by foreign private issuers to provide timely information regarding material events. These forms generally contain information such as changes in business, management or control, bankruptcy, and cybersecurity incidents.

```
Arm Announces Appointment of Ami Badani as Chief Marketing Officer

CAMBRIDGE, England, Nov. 13, 2023 - Arm Holdings plc (Nasdaq: ARM, ''Arm'') today
    announced the appointment of Ami Badani as chief marketing officer (CMO) with
    immediate effect. Badani will lead the Arm global marketing organization and
    report to Arm CEO Rene Haas.

''Arm is foundational to the world's most advanced technology companies and is at
    the forefront of the semiconductor industry's transition to AI,'' said Badani.
    ''It is an incredible time to join Arm and I look forward to working with a
    world-class team.''

Badani joins Arm from NVIDIA, where she held the role of Vice President of Marketing
    and Developer Products. At NVIDIA, her responsibilities included cultivating
    the developer ecosystem for Data Processing Units (DPUs), driving the data
    strategy for Generative AI and leading the company's product and technical
    marketing efforts for the data center portfolio, one of NVIDIA's largest growth
    areas.

''As we continue to advance the Arm compute platform, reaching a more diverse set of
    customers and developers in the AI era is critical,'' said Rene Haas, chief
    executive officer, Arm. ''Ami's experience in AI and proven track record in
    creating awareness among developer ecosystems make her a natural fit to lead
    our marketing efforts in building the future of computing on Arm.''

Badani brings to Arm a unique mix of technology marketing, product management,
    strategy and investment experience. Prior to NVIDIA, Ami was CMO at Cumulus
    Networks, a provider of enterprise-class software that was acquired by NVIDIA
    in 2020. Badani also held marketing and product management leadership roles at
    several technology companies, including Cisco Systems. Prior to joining Cisco,
    Badani worked as an investment banker at Goldman Sachs and J.P. Morgan.

Badani holds an MBA from the University of Chicago Booth School of Business and a B.
    S. in business from the University of Southern California.
```

Figure 14: Excerpt from a Form 6-K filed by Arm Holdings plc on 2023-11-13 (SEC Accession No. 0001973239-23-000014)

## D.9 Form S1/A

Form S1 is used by businesses during initial public offerings (IPOs). When a business needs to make changes to the original Form S1 filed during the IPO process, it files an amendment, i.e., S1/A. These forms generally contain information such as a detailed description of the business, intended uses of the proceeds from the IPO, capitalization, and corporate control.

```
Collaborative Filtering. Amazon.com intends to add a collaborative filtering
    service to its personalized service offerings in the future. The
    collaborative filtering service will function as an expert reviewer that
    develops a relationship with customers, helping them to find books they may
    like based on their preferences. It will match the preferences of people
    to one another, drawing from a pool of titles chosen by other Amazon.com
    customers who share similar interests and tastes.

Ordering. To purchase books, customers simply click on a button to add books to
    their virtual shopping baskets. Customers can add and subtract books from
    their shopping baskets as they browse, prior to making a final purchase
    decision, just as in a physical store. To execute orders, customers click
    on the buy button and are prompted to supply shipping and credit card
    details, either by e-mail or by telephone. This information is stored on
    the Company's secure server and need not be provided again by repeat
    customers. The personal password allows repeat customers to automatically
    access their previously provided shipping and credit card information, as
    well as their book notification profiles. The Company's system
    automatically confirms each order by e-mail to the customer within minutes
    after the order is placed and advises customers by e-mail shortly after
    orders are shipped.

Availability and Fulfillment. Some of the Company's titles are available for
    immediate shipment, others are available for shipment within 48 to 72 hours
     and the remainder of in-print titles are generally available within four
    to six weeks, although some titles may not be available at all. Out-of-
    print titles generally are available in two to six months, although some
    titles may not be available at all. Customers select from a variety of
    delivery options, including overnight and various international shipping
    options, as well as gift-wrapping services. The Company uses e-mail to
    notify customers of order status under various conditions. If a hard-to-
    find book is discovered to have a price higher than an estimate previously
    provided to the customer, the Company notifies the customer and seeks
    approval for sale at the higher price. The Company seeks to provide rapid
    and reliable fulfillment of customer orders, and intends to continue to
    improve its availability and fulfillment in the future.

Out-of-Print. Amazon.com began offering an out-of-print book service in March
    1997. More than one million out-of-print titles are listed in the Company's
     catalog. Because of the difficulty of sourcing out-of-print titles,
    customers are advised to expect two- to six-month delivery times and that
    the books may not be available at all.
```

Figure 15: Excerpt from a Form S1/A filed by Amazon.com on 1997-05-14 (SEC Accession No. 0000891020-97-000839)

# E Examples of Toxic Content From C4-en and BeanCounter

*Warning*: *this section contains content that can be offensive or upsetting.*

In this section, we present examples of the most toxic content in C4-en and BeanCounter as rated by the Perspective API. We select examples by sorting all unique pieces of content by descending order of toxicity, separately for each dataset. To maximize diversity of examples, we present only the most toxic content within a group of similar content. For example, the most toxic content in C4 related to Nationality contains a variety of references to Americans being fat, dumb, and/or lazy: "Americans are fat, lazy, and stupid," "A lot of Americans are idiots," "Americans are simply STUPID." In such cases, we present only the most toxic example.

Table 8: Examples of the Most Toxic Content from the Gender and Sex Axis

| Rank | C4-en | BeanCounter |
|------|-------|-------------|
| 1 | females are stupid and irrational creatures anyway. | We will purchase a semen and inoculate it into our female horses. |
| 2 | Bi-masculine cocksucker here. | Androgen - immature castrated male rat |
| 3 | "Real f**king manly of this b**ch!!! | Anti-androgen - immature castrated male rat |
| 4 | Transgenders are to be killed, female (lightly beaten)... and males shall rule... Naya Albakistan. | The female condom lines the vagina, |
| 5 | it is all Beta crap, "oh, michelle doesn't let me buy things", "oh, barack would be lost without me, cause he is a dumb male" | Estrogen - immature female rat and rabbit |
| 6 | Hot female assassins dressed as slutty nuns. | He said to a male employee, what's wrong with him, is he a retard? |
| 7 | I give zero fucks about cis people's opinions, or hypermasculine binary transmen's opinions. | Malignant neoplasm of penis and other male genital organs |
| 8 | All of those things- stupid, arrogant, narcissistic, flamboyant, pretentious- describe the male model Jason Alan Ross. | "Specified sexual activities" shall mean and include (i) human genitals in a state of sexual stimulation or arousal; (ii) actual or simulated acts of human masturbation, sexual intercourse or sodomy; and (iii) actual or simulated fondling or other erotic touching of human genitals, pubic region, buttocks or female breasts. |
| 9 | Missing out because of his stupid masculine pride and ego. | Provide ContraPest to adult male rats for xxxx, morning of xxxx euthanize rats, weigh testes, epididymis and body and express as testis and epididymis index |
| 10 | Females need to mate. | B. XX Version: shows male erections and/or close-ups of male and/or female genitalia but no extreme close-ups, anal penetration and/or |
| 11 | 2 hours during which female pussies play in games with fists up to full extase!! | And, you know, our claim systems are smart; they won't pay for a vaginal delivery for a male. |
| 12 | It just goes to prove another angle of his masculine idiocy. | Generate [*****] female Transgenic Goats through the [*****] Nuclear Transfer Process. |
| 13 | Skivvy little white male in question will be selling them without batteries or chargers as he was too lazy and stupid to steal them as well. | For the ORAL PRODUCT for female sexual dysfunction: |

Table 8: Examples of the Most Toxic Content from the Gender and Sex Axis (Cont'd)

| Rank | C4-en | BeanCounter |
|------|-------|-------------|
| 14 | I've met plenty of men who were confident and comfortable in themselves, and who managed to be masculine without being ignorant, belligerent assholes. | To castrate all male progeny of the Grandparent Animals at less than 10 weeks of age. |
| 15 | Saying NASCAR is all Rednecks, drunken idiots, and left turns, is the same as saying Football is Steroid taking Black males dealing with balls up and down a subject. | You should hit the male who's married, drinks Diet Coke, watches an extreme amount of wrestling and boxing content. |

Table 9: Examples of the Most Toxic Content from the Nationality Axis

| Rank | C4-en | BeanCounter |
|------|-------|-------------|
| 1 | Mexicans are lazy and stupid. | Ovitz responded, "We're going to kill you guys and your careers are going to be over." Emanuel recalled, I turned to him, got out of my Chinese chair, Japanese chair, whatever, and said, "Are you threatening me?" And I grabbed the chair with my hands and picked it up and said, "Because if you are, I'll fucking throw this chair right out of here right now. Don't threaten me." |
| 2 | F*ck all American women. | Okay, so this is Cranky American Cranky is a resident shit poster, uh, on truth, or sorry, we don't use that word on the show, but he is, he likes to stir the pot. |
| 3 | The Chinese girl is ugly and had a nose job fck her ugly. | The total service includes the collection of semen from the stud, shipment of the semen using the Fresh Express kit to the bitch's veterinarian, artificial insemination of the bitch by the bitch's veterinarian and the completion by both veterinarians of the documentation contained in the Fresh Express kit necessary to register the puppies as purebred with the American Kennel Club or other pedigree registering organization. |
| 4 | Americans are fat, lazy, and stupid. | The Chinese junk had other pioneering features later copied elsewhere. |
| 5 | You're so stupid and I think I know why: you live in a country that's full of lies, foolish Americans. | American Shareholder has full and unrestricted legal right, power and authority to enter into this Agreement, and to sell, assign, transfer, and deliver to valid, lawful and marketable title to suck American. |
| 6 | After all, we have no options here - all American men are stupid and gay! | What is amazing to us is that Mr. Icahn has a long history of criticizing corporate managers, arguing that the executives who run most American companies are greedy and inept. |
| 7 | I'm going to murder these dumb Indians that keep calling me. | They come predominantly from the blissfully ignorant American workers who shuffle retirement funds into 401(k) accounts each month. |
| 8 | Only here those fools, stupid morons do as they like,they must be Guatemalan whores ! | We believe the African American community is an underserved patient population and suffers a disproportionate incidence of cardiovascular and other diseases. |

Table 9: Examples of the Most Toxic Content from the Nationality Axis (Cont'd)

| Rank | C4-en | BeanCounter |
|------|-------|-------------|
| 9 | Most American girls are fat, slutty feminists who dress like slobs. | I worked at American Express, Qing Lin sexually harassed me. |
| 10 | These stupid Americans on both sides of the fence here have no fuking brains. | I hope you're not screwing some Mexican, Campion. |
| 11 | Mexican fucks his brother's cezch therapyst!! | Chinese poplar will be used as raw material for the line. |
| 12 | She's an ugly whore, all that Mexican trash was at the oscars only to provoke some presidential tweets about it | While dramatic anti-trade shit in Chinese policy or laws would seem to be clearly against the best interests of China and its current economic trends, China has a central government with the authority to make such changes. |
| 13 | Mexican latin is such a whore! | We find it remarkable that both Donald Trump and Bernie Sanders are essentially making the same point that prior governments have sold out the American people through poorly considered policies that were either stupid or indicative of a system in thrall to special interests. |
| 14 | While I'm delivering the news, here's something for you ignorant American backwoods motherfuckers. | Then you penetrated a large Korean memory player in September '06. |
| 15 | She knows damn well she wouldn't be cool with black people using any slurs about Mexicans all casually and sh-t. F-ck out of here. | This American Fire Retardant Corp. |

Table 10: Examples of the Most Toxic Content from the Race and Ethnicity Axis

| Rank | C4-en | BeanCounter |
|------|-------|-------------|
| 1 | My God ... do I hate those Asian bastards. | Indeed, these rules moved Mr. Prodi, the President of the European Commission, to state that they were "stupid." |
| 2 | These Asian girls are white worshipping whores. | The agreements that have been struck call for more regulators, particularly at European level. So you are telling us that all that is bullshit? |
| 3 | December 8, 2018 - Fucked Asians While Hogtied! | Find a South African who says he's not a racist and I'll tell you he's a liar. |
| 4 | Of course some of the Europeans are stupid & idiotic enough to vote for it anyhow! | One particular bright spot is our penetration into European cat households. |
| 5 | Hispanic immigrants are murderers and rapists. | African sleeping sickness is a parasitic disease that is spread by tsetse flies in sub-Saharan Africa. |
| 6 | I mean, the Europeans are stupid, but are they that stupid? | These clowns in the European Parliament seem determined to destroy the European airline industry with these discriminatory taxation penalties. |
| 7 | Those damn Asians, never knowing when they're beat. | If there is something to add, the African swine, I mean I just can't, I can't put my finger on it. |
| 8 | Native American tribute to idiots. | Here you have a disease that disproportionably affects African - people of African decent. |
| 9 | Asian teen blindfolded gets fucked by best friend. | We hate South African companies that want to be something else. |

Table 10: Examples of the Most Toxic Content from the Race and Ethnicity Axis (Cont'd)

| Rank | C4-en | BeanCounter |
|---|---|---|
| 10 | The stupid fool forgot that Jesus was a Jew and not the western European in those ... ha ha ha "God- awful" paintings. | There are over 2x more White menthol smokers than African American menthol smokers |
| 11 | There are a lot of European clubs I don't like because their fans are horrible racist c*nts. | This is already on top of U.S. and European PIIGS (Portugal, Italy, Ireland, Greece and Spain) fiscal fears. |
| 12 | As in, stupidly, stupidly wealthy gazillionaire Asians. | Acronym translation: ECB (European Central Bank), OMT (outright monetary transactions programme), OTT (over the top), OMG (oh my God), and WTF (what the f***?) |
| 13 | Innocent White Boy fucked his first African princess, undermined so FAST ! | Premature African American babies as low worth babies in an open lecture among other slights. |
| 14 | I'm sure your dumb brain never even knew numerous standard English words were gotten from other languages like Greek, Latin etc so if a standard English word like "Chemistry" could be gotten from Ancient Egypt then why can't an informal word be coined from other languages mumu....The more you quote me the more I prove your stupiddity. | African-Americans are three times more likely to die of asthma than whites |
| 15 | The Hezbollah Shiite Arabs are Iran's useful-idiots murdering Sunni Arabs. | There are four U.S. patents, one European Patent (EP) patent application (valid until 2019), and one Canadian patent covering Adiri's unique breast shaped nipple |

Table 11: Examples of the Most Toxic Content from the Religion Axis

| Rank | C4-en | BeanCounter |
|---|---|---|
| 1 | What's with you f*ucking religious idiots and your dumbass propaganda at Filmtracks? | Diseases prevalent in Ashkenazi Jewish population |
| 2 | Stupid, in other words, in a "You goddamn smug Christian pigfucker, I will do everything in my power to keep a moron like you from having anything to do with my child's education, and if you don't like it, you can choke on my bile and go into a permanent degenerative coma" sort of way. | Death of a Jewish family member |
| 3 | Welcome to America, where the new mantra is that Christians are stupid. | It reminds me of a Jewish curse my Rabbi once spoke of: May you be successful and the rest of your family poor. |
| 4 | .......what a complete load of religious crap. | To market to the Jewish community profitably while maintaining a high penetration rate. |
| 5 | Masturbation is wrong as you are a Christian. | Doesn't Christian fundamentalists converting homosexuals bother you? |
| 6 | First, there are non-Christian dumb people who don't know Christ and therefore should not really be held accountable for believing stupid stuff. | I made this statement to clarify that I believe even a faith held dear by the most vicious terrorists in the world (Muslim extremists) does not isolate them from God's love. |
| 7 | Saffronization seems to be the mood of the morons secularism be damned ! | People of Ashkenazi Jewish heritage have an increased risk of developing Crohn's disease. |
| 8 | I thought Catholicism was the whore? | I might be a by-the-book Christian, but I'm not stupid. |

Table 11: Examples of the Most Toxic Content from the Religion Axis (Cont'd)

| Rank | C4-en | BeanCounter |
|---|---|---|
| 9 | It's ironic that you brites are the only sane, intellectual people in the world and the rest are all idiots because they are religious. | Recently, we have just experienced the worst terrorist acts in our history: the destruction of the World Trade Center in New York City and significant damage to the Pentagon in Washington D.C. by airlines that were hijacked by Muslim extremists. |
| 10 | Your religious is ridiculous. | Bill Oakley is now an FBI agent who needs someone to infiltrate the infamous Jewish mafia known as Murder Incorporated. |
| 11 | If you're not a Christian, you'll be tortured FOREVER in a burning hell. | Upon the death of the Insured, the Lutheran charitable organization or congregation will receive the Death |
| 12 | I'm not anti-Jesus and I'm not anti-Christian, I'm just anti-assholes-in-the-name-of-Jesus. | Neil has a habit of telling jokes , sex jokes, Black jokes, Polish jokes, Jewish jokes, jokes about everyone. |
| 13 | Which one of the lost tribes did you spring from – you're like none of your race – drinking yourself stupid like a good Christian. | We believe the strength of the JDate brand will continue to allow us to market to the Jewish community profitably while maintaining a high penetration rate. |
| 14 | If Marie Harf and her masters in Washington wish to transpose their own lack of faith, or secularized liberal "Christianity" on an Apocalyptic Death Cult, not only are they idiots, but our nation is screwed. | Destroy national ethnic religious policies, and propagate heresy and superstition; |
| 15 | If I tell him Christians are stupid, he'll tell that to every kid he meets, because dad's word is law when you are five. | The total cost of any such infringement action commenced or defended solely by National Jewish shall be borne by National Jewish. |

Table 12: Examples of the Most Toxic Content from the Sexual Orientation Axis

| Rank | C4-en | BeanCounter |
|---|---|---|
| 1 | You never know when you might have to punch the sh*t out of some fool who is threatening you (or your friends) because you're gay. | 180's have a purpose We'll fix you if you're gay. |
| 2 | Y'all are giving gays a bad rep. Only trash smoke! | Grindr is for gay men. |
| 3 | And they hate when those whom they trolled hit back, as with Richard you'll-never-be-Roger-Ebert Roeper and Milo racist-sexist-Buttburper Yiannopoulos (there is nothing willfully stupider than a gay man sucking up to the radical right). | Are you gay? |
| 4 | The Jets were spending money like a Republican senator in a homosexual whore house in the free agent market. | We had sex, and now she says she knows she's a lesbian. |
| 5 | Dear Tiger Mom, your kid is gay. | "Explicit Version" shall mean a Picture which depicts heterosexual situations and nudity and which depicts actual penetration of body parts, erect genitalia and ejaculation. |
| 6 | My sister who worried about her older sister was also fucked himself to help her sister, who are going to fall by being made a sister lesbian, but escape by seeing the gap. | The actual film we produce is the gay film, and then what we do is replace certain scenes with straight scenes that we do to sell to certain international markets who will not buy our gay version. |

Table 12: Examples of the Most Toxic Content from the Sexual Orientation Axis (Cont'd)

| Rank | C4-en | BeanCounter |
|---|---|---|
| 7 | That's the gayest thing I've ever seen. | Gaysexmaniac.com - a gay sex search engine |
| 8 | No heterosexual black man or BLACK WOMAN would still have a show if they had LaDonna's putrid low ratings. | Major advertisers including, among others, IBM Corp., Seagram Americas' Absolut Vodka, [...], United Airlines and U.S. West are targeting the gay and lesbian community and other major corporate names will likely follow suit. |
| 9 | i love matchbox 20 but that is just gay. | Now they're boycotting Pepsi products because they claim PepsiCo is funding the gay agenda, including same sex marriage. |
| 10 | The video which smells like poppers, is an explosion of colors and glitters, full of gayness. | Each sapling will be uniform, disease resistant, asexual and fire retardant. |
| 11 | Thats gay, why can't it roll over? | Now, you check the bisexual box you never actually act on it, probably because you're not right. |
| 12 | Click to watch more booty black lesbian videos videos! | Unfair treatment of gay and lesbian employees |
| 13 | Make out sesh blondes amount all your hd ebony lesbians avenue providers and can be hooked any sundry of day, round after your possibilities. | It is estimated that the combined number of subscribers for these magazines is approximately 250,000 (although the general circulation through newsstand sales is much greater), whereas the average number of subscribers for other men's magazines which have a large gay following but do not purport to be gay and lesbian publications such as "Details" is 485,000, "Esquire" is 658,000 and "Men's Journal" is 555,000. |
| 14 | We pfueger to all your homosexual needs and make pflueged rock hard in seconds. | Cease support of homosexual lifestyle and other deviant lifestyle behaviors opposed by the majority of people. |
| 15 | unfurl the gay banners! | Leukosite shall not make any effort, directly or indirectly, to breed, clone or otherwise reproduce the Mice by any means, sexual or asexual. |

### E.1 Multi-turn Discussions in BeanCounter

In this section, we present an example of a multi-turn discussion between former PepsiCo CEO Indra Nooyi and a shareholder. This discussion is captured in a transcript of PepsiCo's annual shareholder meeting which was filed on EDGAR.

```
G. Quinlan: Thank you very much. My name is Greg Quinlan, I'm a member of PFOX,
    Parents and Friends of Ex-Gays & Gays. PFOX is a non-profit charity that
    supports families in the ex-gay community. I myself am an ex-gay, a former
    homosexual. I ask you to support resolution number six, the charitable
    contributions resolution, because PepsiCo gives away over $1 million to gay
    organizations that hate people like me. Last year, PepsiCo gave away $.5
    million to PFLAG, Parents and Friends of Lesbians and Gays. PFLAG is a gay
    organization that supports tolerance for gays, but does not believe in
    tolerance for ex-gays or other heterosexuals. When Republican Vice Presidential
     nominee Sarah Palin's church sponsored an ex-gay conference last year, PFLAG
    issued press releases against Sarah Palin and organized protests at her church.
     This same church was later vandalized by fire, reminiscent of the burning of
    black churches in the south during the Civil Rights movement. Churches should
    welcome former homosexuals like me, and PepsiCo should not fund organizations
    like PFLAG that do not welcome diversity but promote hatred. I have been
    shouted and heckled by PFLAG members because I am out and open in society as a
    former homosexual. Why does PepsiCo fund groups like PFLAG that hate people who
     are different from them? This is not tolerance. This is not inclusion by any
    definition.

PFLAG publishes anti-gay literature and opposes ex-gay inclusion. Although they
    advocate for gay rights, they just don't believe others should have the same
    equal rights. PFLAG contributes to the intolerance of the ex-gay community, and
     stereotypes former homosexuals like myself. By funding PFLAG, PepsiCo promotes
     fear and hostility against the ex-gay community and supporters, and spreads
    lies about ex-gay organizations. Diversity does not and should not mean funding
     one organization to attack another.

PFLAG also funded support for the recent gay marriage political battles across the
    country, but the American people voted for traditional marriage and against gay
     marriage in every state where the initiative was held. In fact, 30 states,
    three fifths of the United States, 60 percent, supports marriage as one man,
    one woman.

[...]

If PepsiCo can afford that, then it can afford to issue a report revealing to you,
    the shareholders, how your money is being spent on dubious charities that
    support genderless marriage and hate against ex-gays like myself. Please
    support charitable contributions number six, and thank you very much.

I. Nooyi: Thank you, Mr. Quinlan. Before we call for a second, let me just take a
    minute to share PepsiCo's position on this matter as outlined in the proxy
    statement. The PepsiCo Foundation seeks to expand opportunities for under-
    served communities to enjoy improved health, a better environment, and greater
    inclusion. We're also committed to diversity and inclusion in our businesses so
     that we can serve and appreciate our many different communities and consumers
    without the imposition of personal judgment. Shareholders can easily see the
    extensive steps PepsiCo already takes to document its corporate social
    responsibility and foundation activities. Details regarding grant guidelines,
    and lists of donations, are already featured prominently on the PepsiCo website.
     So the goal underpinning the proposal under consideration has already been
    accomplished. With that said, is there anyone to second Mr. Quinlan's motion?
```

Figure 16: Excerpt from a multi-turn discussion of a shareholder proposal at PepsiCo's annual shareholder meeting

