# OpenReview forum: "BeanCounter: A low-toxicity, large-scale, and open dataset of business-oriented text"
_NeurIPS.cc/2024/Datasets_and_Benchmarks_Track — NeurIPS 2024 Track Datasets and Benchmarks Poster_

### Official Review · Reviewer_Hg9p · 2024-06-21
**Review of BeanCounter - a large natural language dataset from financial filings**

**Rating:** 8
**Confidence:** 4
**Clarity:** I did not find any errors - the paper…

**Review:**

The dataset that the authors have prepared is a well thought out processing of an extremely large corpus of public domain data that is not part of the common webcrawl datasets used to build most large language models. The report is well written and covers the important details regarding how the text was prepared.

In additional to presenting this prepared dataset, the authors perform a variety of analyses that compare this resource to standard resource and show that it is demonstrably better in terms of aggregate statistics. They also go to some length to demonstrate that this will improve models, although this portion is somewhat sparse as only two models are considered. This weakness is not disqualifying, however.

While I believe this dataset and paper are appealing and a solid contribution, I do think that the authors have not considered the possible downsides to this data and the potential for alignment problems. The time periods covered by the data collected include some of the largest regulatory failures in financial history, including the Enron scandal, the Madoff investment scandal, the Savings and Loan scandal, and many others. Despite the "highly regulated environment" that the authors contend makes this text better than that sourced from the web, the systematic way that businesses are able to exaggerate upside benefits and minimize downside risks means that training on this text could create systems with misaligned loss aversion characteristics. While I do not believe that these risks are disqualifying, I think they should be acknowledged.

**Strengths:**

The authors have identified a substantial and potentially very useful corpus comparable to the webcrawls used to train LLMs.

The comparisons to estimate potential gender, demographic, and tendency towards toxicity are all exemplary.

The experimental coverage is modest, but sufficient to demonstrate the real potential of this resource.

**Additional Feedback:**

None

**Correctness:**

The sourced data and processing all seem straightforward and appropriate.

The bias analysis seems sound, and the fine tuning experiments also seem quite reasonable.

**Documentation:**

Well covered.

**Ethics:**

No concerns.

**Limitations:**

Businesses often have financial incentives to produce content that is in compliance with regulations but otherwise deceptive. Language models trained upon business rationales could have subtle and difficult to detect alignment problems.

Business writing may have other biases beyond those tested by the authors.

**Opportunities For Improvement:**

Additional testing with other large language models would strengthen the case that this data is beneficial compared to web crawls.

More attention to the potential ways that business communication can be manipulated and/or deceptive to both potential investors and regulators.

On this last matter, some analysis of the difference between SEC content and web content might open new avenues of inquiry. (For example, if models trained on SEC have a tendency to generate more positive sentiment towards descriptions of opportunities or products, this would itself offer research opportunities.)

**Relation To Prior Work:**

Comparisons to webcrawls is fine. I am not aware of similar work beyond what is cited.

**Summary And Contributions:**

The authors have prepared a substantially large data set, comparable in size to web crawl data currently used to train large language models. The authors make several comparisons concerning the aggregate characteristics of the source text and show that it may be more balanced on gender and sexual orientation, and that these characteristics may make help produce models with less bias. They also show that the coverage regarding financial concepts may itself make models trained with this data have higher performance on business related tasks.

---

> ### Author Rebuttal · Authors · 2024-08-17
>
> We appreciate your helpful review and believe the feedback has greatly improved our paper. Attached is an updated draft. We reproduced your comments below alongside the changes we made in response. We look forward to your thoughts on the update, thank you!
>
> **Feedback:** Additional testing with other large language models would strengthen the case that this data is beneficial compared to web crawls.
>
> **Response:** Unfortunately, we do not have a large cluster of GPUs so training models on a larger scale isn't feasible for us. To address your feedback, we have updated the paper to discuss this limitation of the study explicitly in Section 5. Limitations and suggested some directions for future research such as varying the mix of BeanCounter with other pre-training datasets.
>
> **Feedback:**
> - More attention to the potential ways that business communication can be manipulated and/or deceptive to both potential investors and regulators.
> - The time periods covered by the data collected include some of the largest regulatory failures in financial history, including the Enron scandal, the Madoff investment scandal, the Savings and Loan scandal, and many others. Despite the "highly regulated environment" that the authors contend makes this text better than that sourced from the web, the systematic way that businesses are able to exaggerate upside benefits and minimize downside risks means that training on this text could create systems with misaligned loss aversion characteristics. While I do not believe that these risks are disqualifying, I think they should be acknowledged.
> - Businesses often have financial incentives to produce content that is in compliance with regulations but otherwise deceptive. Language models trained upon business rationales could have subtle and difficult to detect alignment problems.
>
> **Response:** We felt that these three excellent points were related and we made changes in response to them jointly.
> 1. Businesses will always have incentives and this needs to be acknowledged. We now discuss this in the Limitations section in the paragraph "Deceptive Content"
> 2. Removing as much known deceptive content from BeanCounter as possible will mitigate but not solve this limitation. Along these lines, we used data from the SEC's enforcement actions against firms that committed fraud to identify disclosures which were filed on EDGAR during periods of fraud. For example, [WorldCom](https://www.sec.gov/enforcement-litigation/litigation-releases/lr-18219#:~:text=The%20Commission%20has%20alleged%20that,17829.). We identified 286 firms which the SEC took action against, spanning 757 fraudulent firm-years. We then removed filings during these firm-years from the BeanCounter "final" split.
> 3. We created a new "fraud" split of the data which contains the filings identified in Step 2 above. We find that this contains roughly 300M tokens or about 0.2% of the original dataset. We hope that this split may be useful for future research on detecting misleading content.
>
> We are interested in your thoughts on this since the data on frauds is limited to those firms conducting deceptive activities which were detected by the SEC, but feel like it is better than not doing any filtering.
>
> **Feedback:** On this last matter, some analysis of the difference between SEC content and web content might open new avenues of inquiry. (For example, if models trained on SEC have a tendency to generate more positive sentiment towards descriptions of opportunities or products, this would itself offer research opportunities.)
>
> **Response:** This is a very interesting idea – we also wanted to understand whether BeanCounter has a positive sentiment bias towards products. As preliminary exploration, we randomly sampled 1.1 billion tokens from each dataset and extracted sentences where there is a mention of a product with spaCy’s NER identifier. We identified 638,170 and 750,561 product sentences from BeanCounter and C4-en respectively. We used the twitter-roberta-base-sentiment-latest Huggingface model as the pipeline for sentiment evaluation. Of the BeanCounter product sentences, 95.74% were classified as neutral, 2.82% as negative and 1.44% as positive. For the C4-en product sentences, 70.91% were classified as neutral, 22.77% as positive and 6.32% as negative.
>
> As one can see from the analysis, BeanCounter product sentences may in general be more “emotionally-restrained” compared to C4-en. This is possibly a product of business disclosure writing conventions.
>
> There are several important caveats to these numbers. Firstly, we found that most off-the-shelf NER models are not very good at identifying products in a sentence; due to time constraint, we could not re-train an NER model specifically for product recognition. Additionally, the text included in BeanCounter is stylistically and content-wise very different from most web-crawled datasets. This difference may contribute to the inaccuracy in product recognition from NER models that are presumably trained on web-crawled corpuses.
>
> As of right now, we have not incorporated this analysis into the paper due to concerns with the accuracy of NER tagging. But if you think it is worth including in an appendix, we are happy to add it.

---

> > ### Comment · Reviewer_Hg9p · 2024-08-17
> > **Acknowledgement**
> >
> > Thanks for this response - I think the inclusion of a section and slice on the general topic of fraud adds what was a missing critical perspective. This goes a long way against my concerns. The additional testing is intriguing but not sufficient for me to significantly change my review. I still think this is a valuable contribution and encourage its acceptance.

---

### Official Review · Reviewer_M1Cw · 2024-07-29
**The authors construct a large dataset of over 100B tokens based on public business data. They also provide some analysis of this dataset and demonstrate its utility by continuing pre-training 2 'small' models on this data. Overall, this work seems sufficient for publication.**

**Rating:** 6
**Confidence:** 5
**Correctness:** Analysis and methodology seems correct.

**Review:**

Please see "Strengths" and "Opprotunities for Improvement" for more details. Overall, I belive this paper is adequate for publication. The size of the dataset as well as the 2 unique pros the authors point out (domain specific utility and lack of toxicity / bias relative to standard internet-sourced datasets) seem sufficient to essentially guarantee this dataset will be useful for some purposes.

However, I strongly urge the authors to do some more analysis prior to publication to help readers understand the utility of this dataset and to properly understand its limitations. Some of the analyses that I think would be most useful are highlighted in the "Opprotunities for Improvement" section.

**Strengths:**

- The dataset size is quite large and, as the authors point on, on the scale required for use in pre-training. The authors also have a reasonable pipeline for cleaning and deduping the data that is presented in adequate detail.
- The authors provide some basic analysis of what is in the dataset, and in particular, demonstrate the diminshed amount of toxic data in the dataset.
- The authors demonstrate models pre-trained with their data do, in fact, have higher performance on business and financial-related tasks as well as lower bias and toxicity according to standard measures.

Overall, I belive this is adequate for publication.

**Additional Feedback:**

Minor typos:
- In Figure 5 description, "after continually pre-trained on BeanCounter" should be "after continued pre-training on BeanCounter"
- As mentioned in "Opportunities for Improvement," Figure 5 also suggests there is also a decrease for "jewish," which should be mentioned in the text in line 250.

**Clarity:**

Writing is clear and easy to understand. In particular, the methodlogy is easy to understand.

**Documentation:**

The dataset creation and methodology is well documented and easy to understand.

**Ethics:**

No ethics concerns.

**Limitations:**

Limitations are addressed. See comment in "Opportunities for Improvement" about model evaluations.

**Opportunities For Improvement:**

- I would like more examples to be included in the appendix. For example, according to the toxicity classifer, what are examples of the most toxic data in the BeanCounter dataset? Are these examples similar to toxic data in, for example, C4? I would also include in the Appendix a (potentially truncated) example of some of the most common types of documents included in the BeanCounter dataset, so that readers can get a rough sense of the data without looking at sources external to the paper.
- Table 3 provides a good overview of how BeanCounter has less toxic data, and I think the metrics chosen for (a) and (b) are appropriate. In the appendix, I suggest the authors include the actual average toxicity scores computed as well.
- Table 4 suggests only modest gains in performance and reduction in toxicity. I would like to see additional studies on more models with larger sizes etc. Of course, this requires quite a bit of computational budge and time, so this is not something I expect or think is required for an acceptable paper. However, I do think this should be (briefly) discussed as a limitation.
- Figure 5 suggests there is also a decrease for "jewish," which is not mentioned in the text in line 250. More importantly, the authors should do more analysis to attempt to understand *why* there is an increase in safety score after continued pre-training. Is it just random chance, or is it a fundamental bias in either the BeanCounter dataset or the evaluation method? Given that the authors are claiming BeanCounter can be used to reduce bias, it is important to understand its limitations and when it can / cannot help with this.
- Section 4.2 analysis needs to be paired with some standard LLM benchmarks. Being able to decrease toxicity is great, but to assess the utility of BeanCounter, we also need to know if this reduction in toxicity is accompanied by any changes to the performance of the model on other tasks it may be used for. Section 4.1 suggests there is an improvement for a specific domain of tasks (which is not surprising), but if BeanCounter is more broadly useful for toxicity reduction in general, we need to see the analysis that toxicity reduction does not impact performance on generic tasks.

**Relation To Prior Work:**

Prior work is adequately discussed.

**Summary And Contributions:**

The authors construct a large dataset of over 100B tokens based on public business data from EDGAR filings. While others have previously used EDGAR filings to construct such datasets, the author's dataset is more comprehensive and roughly 10x larger than any previously released datasets based on business / financial data.

The authors then provide basic analysis of this data describing where (which companies and which years) most of the tokens come from, the types of documents the tokens come from, and demographic and toxicity content of the data (i.e., prevalence of different pronouns, toxitity of content, etc.).

The authors then demonstrate two potential use cases for their dataset: (1) as a domain-specific data for pre-training models (for business or financial use cases) and (2) as a source for less biased and toxic data for reducing toxicity and bias in models. To demonstrate these use cases, the authors continue pre-training 2 different "small" models. They then show across a number of standard benchmarks the effects of the continued pre-training.

---

> ### Author Rebuttal · Authors · 2024-08-17
>
> Thank you for the helpful review, we believe the feedback has greatly improved our paper and look forward to your thoughts on the update. We have attached an updated draft and reproduced your comments below alongside the changes we made in response.
>
> **Feedback:** I would like more examples to be included in the appendix. For example, according to the toxicity classifier, what are examples of the most toxic data in the BeanCounter dataset? Are these examples similar to toxic data in, for example, C4?
>
> **Response:** Thank you for the suggestion, this will definitely be helpful to readers. We have now included examples of the most toxic sentences from BeanCounter and C4-en across the 5 axes in Appendix E.
>
> We have also added a brief discussion of the examples to Section 3.4, reproduced here: “In addition to quantitatively exploring differences in toxicity, we also read and compare examples of toxic content from C4-en and BeanCounter. We find that toxic content in BeanCounter tends to come from the discussion of business models which involve breeding animals, medical ailments which affect certain portions of the population at different rates, firms providing examples of actions that would violate their ethics policies, and entities raising money to produce certain artistic works, e.g., movies, which contain content labeled as toxic. In contrast, the toxic content in C4-en tends to be directed at specific groups.”
>
> We think that the toxic content in BC is substantially different than that in C4, but would like to get your thoughts after you have a chance to look at the appendix.
>
> **Feedback:** I would also include in the Appendix a (potentially truncated) example of some of the most common types of documents included in the BeanCounter dataset, so that readers can get a rough sense of the data without looking at sources external to the paper.
>
> **Response:** We agree that this would be helpful and have added a new Appendix, D, in which we present brief descriptions of the 10 most common forms as well as truncated examples of content. These were largely chosen at random, although we thought it was interesting that Amazon, in their 1997 S1/A filing, discussed plans to use IPO funds for implementing collaborative filtering so that example is hand-picked.
>
> **Feedback:** Table 3 provides a good overview of how BeanCounter has less toxic data, and I think the metrics chosen for (a) and (b) are appropriate. In the appendix, I suggest the authors include the actual average toxicity scores computed as well.
>
> **Response:** This is an excellent point and admittedly we do not have access to a large GPU cluster. We appreciate you seeing value in the dataset even though we are unable to run a large model training study. In addition to the studies you mention, we also think it would be interesting to examine the impact of training solely on BeanCounter or incorporating it into pre-training as various percentages of the data mix. We have added discussion of these points to a new Section, 5. Limitations, under the paragraph “Ablation”
>
> **Feedback:** Figure 5 suggests there is also a decrease for "jewish," which is not mentioned in the text in line 250.
>
> **Response**: Thank you for catching this, it has been corrected in L266.
>
> **Feedback:** More importantly, the authors should do more analysis to attempt to understand why there is an increase in safety score after continued pre-training. [...shortened for brevity]
>
> **Response:** We have spent some time thinking about this and reflecting on what we have seen when qualitatively exploring the data. One hypothesis we have is that when toxic content appears in BeanCounter, it tends to be in a context where there is either a civil discussion around the topic or the firm is explaining why some bad behavior is a violation of their ethics policy.
>
> In the spirit of multi-turn prompting, we think examples like this in the training data may lead to some of the improvements we observe. We’d like your thoughts on this because though we do not have any rigorous tests of this hypothesis in the paper and aren’t immediately sure how to approach testing it. But we have added an example of a multi-turn conversation from PepsiCo’s shareholder meeting (see Appendix Section E.1) so that you can see what we mean. We have also added a brief discussion of this hypothesis to the end of the Generation Toxicity section (4.2).
>
> **Feedback:** Section 4.2 analysis needs to be paired with some standard LLM benchmarks. [...shortened for brevity]
>
> **Response:** This is a really good point – we have added table 5 in Appendix B which shows the models’ performance on a suite of common LLM benchmarks. From these results, one can see that the Pythia-1.4b model had similar performance or was slightly better at some of the benchmarks after continued pre-training on BeanCounter. However, the Phi-1.5 model exhibited decreases in performance (namely BBH, MMLU-Pro, GPQA) after continued pre-training. We interpret this as evidence that the impact of continued pre-training on BeanCounter may be model-specific (and specific to the model's mix of pre-training data). We have added discussion of these limitations of our analysis to the limitations section in the paper (5)

---

### Decision · Program_Chairs · 2024-09-26

**Decision:**

Accept (Poster)

**Comment:**

In this paper, the authors introduce BeanCounter, an extensive dataset created from public business filings available through the EDGAR database. This dataset boasts over 100 billion tokens, making it significantly larger than existing similar datasets. The authors analyze BeanCounter in terms of demographics, document types, and potential biases and toxicity present within the data. They also conduct pre-training experiments on smaller models to demonstrate how effective BeanCounter is, showing enhancements in domain-specific tasks and lower levels of bias and toxicity compared to typical datasets gathered from the web.


The reviewers agree that this paper makes a noteworthy contribution to the field, with good potential to enhance the use of domain-specific data in natural language processing tasks.

Strengths:

1.	The scope and thoroughness of BeanCounter provide significant advantages for pre-training models, especially in business and finance.
2.	The authors have given a valuable look at bias and toxicity, showing that BeanCounter contains fewer of these issues compared to datasets sourced from the Internet.
3.	The experiments indicate that BeanCounter effectively improves model performance on finance-related tasks while reducing bias and toxicity, despite some limitations due to computational constraints.
4.	The authors offer a transparent account of how the dataset was constructed and cleaned, which supports reproducibility and builds trust in their findings.

Weakness:
1.	Reviewers suggest conducting further tests with a wider range of models to better support the claims about the dataset's advantages.
2.	The discussion could benefit from addressing the potential risks of bias or misalignments in business communications, referencing past regulatory issues like the Enron scandal.

PS: Because the paper has only received two reviews, the AC has read through the paper carefully, and included his review in the meta review.